# Extreme Value Policy Optimization for Safe Reinforcement Learning

**Shiqing Gao** [1]   **Yihang Zhou** [1]   **Shuai Shao** [1]   **Haoyu Luo** [1]   **Yiheng Bing** [1]   **Jiaxin Ding** [1]   **Luoyi Fu** [1]   **Xinbing Wang** [1]

## Abstract

Ensuring safety is a critical challenge in applying Reinforcement Learning (RL) to real-world scenarios. Constrained Reinforcement Learning (CRL) addresses this by maximizing returns under predefined constraints, typically formulated as the expected cumulative cost. However, expectation-based constraints overlook rare but high-impact extreme value events in the tail distribution, such as black swan incidents, which can lead to severe constraint violations. To address this issue, we propose the Extreme Value policy Optimization (EVO) algorithm, leveraging Extreme Value Theory (EVT) to model and exploit extreme reward and cost samples, reducing constraint violations. EVO introduces an extreme quantile optimization objective to explicitly capture extreme samples in the cost tail distribution. Additionally, we propose an extreme prioritization mechanism during replay, amplifying the learning signal from rare but high-impact extreme samples. Theoretically, we establish upper bounds on expected constraint violations during policy updates, guaranteeing strict constraint satisfaction at a zero-violation quantile level. Further, we demonstrate that EVO achieves a lower probability of constraint violations than expectation-based methods and exhibits lower variance than quantile regression methods. Extensive experiments show that EVO significantly reduces constraint violations during training while maintaining competitive policy performance compared to baselines.

## 1. Introduction

Reinforcement Learning (RL) has achieved remarkable success across various fields such as robot control (Haarnoja

[1]Shanghai Jiao Tong University, Shanghai, China. Correspondence to: Jiaxin Ding <jiaxinding@sjtu.edu.cn>.

*Proceedings of the 42nd International Conference on Machine Learning*, Vancouver, Canada. PMLR 267, 2025. Copyright 2025 by the author(s).

et al., 2018; Xu et al., 2020) and games (Vinyals et al., 2019; Yu et al., 2022a). However, the absence of safety guarantees poses a significant barrier to the deployment of RL in real-world scenarios. To overcome this limitation, Constrained Reinforcement Learning (CRL) (Ding et al., 2020; Stooke et al., 2020; Xu et al., 2021; Yang et al., 2022) aims to optimize policies by maximizing cumulative rewards while satisfying predefined constraints. In safety-critical domains such as autonomous driving and financial trading, safety guarantees are highly susceptible to extreme events, which occur with low probability but can lead to significant consequences (NS et al., 2023). For instance, black swan incidents are especially concerning due to their impact to cause catastrophic outcomes (Masys, 2012).

Most CRL methods evaluate constraints based on the expected sum of costs (Achiam et al., 2017; Stooke et al., 2020; Ding et al., 2021), optimizing policies to ensure the expected cost remains below a predefined threshold. However, this expectation-based evaluation only guarantees compliance on average, neglecting the inherent variability of the cost distribution, particularly in extreme samples arising in the tail. Consequently, even when the expected cost satisfies the constraint, frequent constraint violations can occur due to the presence of these extreme events, as shown in Figure 1a.

To minimize the probability of constraint violations, probabilistic constraint methods (Chow et al., 2018; Hiraoka et al., 2019) optimize policies based on constraint distributions. WCSAC (Yang et al., 2021) employs Distributional RL (Bellemare et al., 2017) with a Gaussian approximation of the cost distribution, computing Conditional Value-at-Risk (CVaR) (Rockafellar & Uryasev, 2002) to guide policy optimization. However, the Gaussian distribution fails to accurately capture the tail behavior, as shown in Figure 1b. QCPO (Jung et al., 2022) introduces quantile-based constraint objective with Value-at-Risk (VaR), deriving approximated policy gradient. However, these methods overlook the critical impact of extreme samples during training.

Extreme samples, characterized by their low probability but high impact (NS et al., 2023), play a crucial role in CRL. Specifically, extreme reward samples provide insights into achieving task objectives, while extreme cost samples highlight critical constraints. Unfortunately, these samples

are naturally scarce, arising infrequently and often requiring risky or expensive exploration to obtain. This scarcity poses challenges in modeling tail distribution, leading to high variance (Urpí et al., 2021). Therefore, accurately modeling tail behavior and effectively exploiting these rare but crucial samples is essential for reducing constraint violations and improving policy performance in CRL.

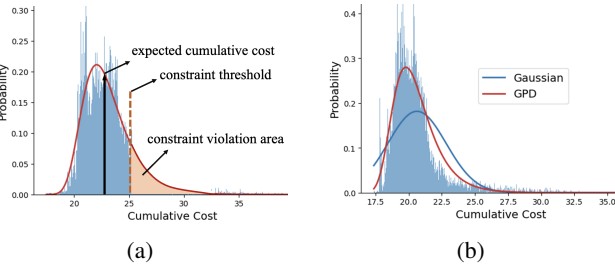

Figure 1: (a) Probability density function of cumulative cost. Even when the expected constraint satisfies the threshold, there remains a high probability of constraint violations. (b) Fitting cumulative cost distribution with GPD and Gaussian, GPD captures tails more accurately.

In this paper, we propose the Extreme Value policy Optimization (EVO) algorithm, which leverages Extreme Value Theory (EVT) (Pickands III, 1975) to model and exploit these samples to enhance policy optimization. To mitigate violations caused by extreme cost samples, we introduce an extreme quantile constraint, explicitly incorporating tail extremes modeled using a Generalized Pareto Distribution (GPD) based on EVT. Additionally, we propose an extreme prioritization mechanism based on quantile levels derived from the GPDs of reward and cost, prioritizing low-probability but high-information samples during experience replay to exploit the learning signals in extreme samples. To address the high variance of extreme samples, we apply an off-policy importance resampling approach, augmenting extreme samples and enhancing tail distribution modeling. Theoretically, we establish an upper bound on expected constraint violations during policy updates in EVO, ensuring strict constraint satisfaction at a zero-violation quantile level. We further prove that EVO achieves lower constraint violation probability compared to expectation-based CRL methods and exhibits lower variance compared to quantile regression methods. Experimental results across multiple environments validate that EVO significantly reduces constraint violations while maintaining strong policy performance. The main contributions of this work are as follows:

1. We propose the EVO algorithm, integrating Extreme Value Theory (EVT) into CRL to model and exploit extreme samples, addressing the limitation in handling rare but high-impact events.

2. We propose an extreme quantile constraint based on Generalized Pareto Distributions (GPDs) and an ex-

treme prioritization mechanism, enhancing learning from extreme samples.

3. We theoretically and empirically demonstrate that EVO achieves lower expected constraint violations, lower violation probability, and reduced variance compared to baselines. We provide the code for EVO in `https://github.com/ShiqingGao/EVO`.

## 2. Related Work

**Expectation Constraint.** The optimization methods in CRL with expectation-based constraints can be categorized into primal-dual and primal approaches. Primal-dual methods (Ding et al., 2021; Ying et al., 2024) convert constrained problems into unconstrained ones using dual variables. NPG-PD (Ding et al., 2020) establishes global convergence with sublinear rates. PID Lagrangian (Stooke et al., 2020) introduces proportional and differential control to mitigate cost overshoot and oscillations. In contrast, primal methods directly optimize constrained problems in the primal space (Yang et al., 2020; Yu et al., 2022b; Gao et al., 2024). CPO (Achiam et al., 2017) enforces performance and constraint violation bounds within a trust region. FOCOPS (Zhang et al., 2020) solves the constraint problem in the nonparametric policy space and projects the policy back to the parametric space. CUP (Yang et al., 2022) provides generalized theoretical guarantees for surrogate functions with generalized advantage estimator (Schulman et al., 2015b). However, these expectation-based methods fail to capture variability in the constraint distribution, resulting in frequent constraint violations. In this paper, we model the tail extremes of constraints using GPD and propose an optimization objective based on extreme quantile to reduce constraint violation probability.

**Probability Constraint.** Probability-based methods (Chow et al., 2018; Hiraoka et al., 2019; Urpí et al., 2021) optimize the probability distribution of constraints to reduce violations. WCSAC (Yang et al., 2021) uses CVaR (Rockafellar & Uryasev, 2002) as a safety measure, approximating the constraint distribution with a Gaussian model. However, the Gaussian approximation fails to capture the tail decay rate, introducing significant bias for small tail probabilities. QCPO (Jung et al., 2022) transforms outage probability constraints into quantile-based ones and proposes an approximate policy gradient. However, these methods neglect the high variance of extreme samples and fail to fully exploit them. EVAC (NS et al., 2023) employs EVT to reduce variance in extreme returns, improving risk aversion in RL, but does not address constraint satisfaction in CRL. In this paper, we propose an extreme quantile constraint optimization objective, leveraging EVT to model extreme samples and reduce variance. Additionally, we

introduce an extreme prioritization mechanism to fully exploit these extreme samples.

## 3. Preliminaries

### 3.1. Constrained Reinforcement Learning

CRL can be modeled as a Constrained Markov Decision Process (CMDP), denoted by a tuple $(S, A, R_f, C_f, P, \rho, \gamma)$, where $S$ is the state space, $A$ is the action space, $R_f : S \times A \to \mathbb{R}$ is the reward function, $P : S \times A \to [0, 1]$ is the transition probability function, $\rho$ is the initial state distribution, and $\gamma \in (0, 1)$ is the discount factor. The cost function $C_f : S \times A \to \mathbb{R}$ maps state-action pairs to costs $c$. $d$ denotes the constraint threshold.

Starting from an initial state $s_0$ sampled from the initial state distribution $\rho$, the agent perceives the state $s_t$ from the environment at each time step $t$, selects an action $a_t$ according to the policy $\pi : S \to A$, receives the reward $r_t$ and cost $c_t$, and transfers to the next state $s_{t+1}$ based on $P(s_{t+1}|s_t, a_t)$. The set of all stationary policies is denoted as $\Pi$. The discounted future state visitation distribution is defined as $d^\pi(s) := (1 - \gamma) \sum_{t=0}^{\infty} \gamma^t P(s_t = s|\pi)$. The value function for a policy $\pi$ is $V_R^\pi(s) := \mathbb{E}_{\tau \sim \pi}[\sum_{t=0}^{\infty} \gamma^t R(s_t, a_t)|s_0 = s]$, and action-value function is $Q_R^\pi(s, a) := \mathbb{E}_{\tau \sim \pi}[\sum_{t=0}^{\infty} \gamma^t R(s_t, a_t)|s_0 = s, a_0 = a]$. The advantage function measures the advantage of action $a$ over the mean value: $A_R^\pi(s, a) := Q_R^\pi(s, a) - V_R^\pi(s)$. The cost value function $V_C^\pi(s)$, cost action value function $Q_C^\pi(s, a)$ and cost advantage function $A_C^\pi(s, a)$ in CMDP can be obtained as in MDP by replacing the reward $r$ with the cost $c$. The expected discounted return $J_R(\pi) := \mathbb{E}_{\tau \sim \pi}[\sum_{t=0}^{\infty} \gamma^t R_f(s_t, a_t)]$, and the expected cumulative discount cost $J_C(\pi) := \mathbb{E}_{\tau \sim \pi}[\sum_{t=0}^{\infty} \gamma^t C_f(s_t, a_t)]$, where $\tau = (s_0, a_0, s_1, a_1, \cdots)$ is the trajectory under $\pi$.

The CRL aims to find an optimal policy by maximizing the expected discount return over the set of feasible policies $\Pi_C := \{\pi \in \Pi : J_C(\pi) \leq d\}$:

$$\arg \max_{\pi \in \Pi} J_R(\pi)$$
$$s.t. \quad J_C(\pi) \leq d \tag{1}$$

### 3.2. Extreme Value Theory

Extreme Value Theory (EVT) is a statistical method for analyzing the distributional properties of extreme events.

**Theorem 3.1.** $X_1, \cdots, X_n$ *denote a sequence of IID random variables with cumulative distribution function $F$, which approaches the Generalized Pareto distribution (GPD) asymptotically. Denote the conditional excess distribution as $F_t(x) = P(X - t \leq x|X > t)$, then:*

$$\lim_{t \to \infty} F_t(x) \to H(x) \tag{2}$$

*where $t$ is a threshold and $H(x)$ denotes the GPD, following:*

$$1 - \left(1 + \frac{\xi x}{\sigma}\right)^{-\frac{1}{\xi}}, \quad \xi \neq 0 \tag{3}$$

Theorem 3.1 describes that the conditional distribution above a threshold $t$ approaches the GPD.

We introduce the necessary notation in this paper. Let $X$ be a random variable with cumulative distribution function $F(x) = P(X \leq x)$. The tail distribution, representing the probability of $X$ exceeding a threshold $x$: $\bar{F}(x) = P(X > x) = 1 - F(x)$. Given a quantile level $\mu$, the quantile $q_\mu$ is defined as the smallest value such that $P(X \leq q_\mu) \geq \mu$ i.e. $P(X > q_\mu) < 1 - \mu$.

## 4. Methodology

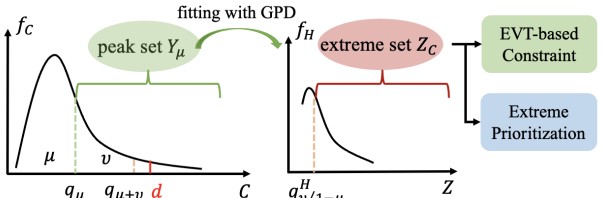

Figure 2: Extreme quantile constraint $q_{\mu+\nu}$ and EVT-based constraint $q_\mu + q_{\frac{H_\nu}{1-\mu}}$. The peak set $Y_\mu$, containing samples exceeding the safety boundary $q_\mu$, is used to fit GPDs. The extreme set $Z_C$, containing samples exceeding the risk boundary $q_\mu + q_{\frac{H_\nu}{1-\mu}}$, is used to compute EVT-based constraints for reducing violations and to calculate extreme priorities for exploiting extreme samples.

In this section, we present the modeling of extreme samples using EVT and derive the constrained optimization objective based on extreme quantile. We then introduce an extreme prioritization mechanism to effectively capture information from extreme samples and enhance extreme value modeling using off-policy samples to mitigate the high variance. Finally, we provide theoretical guarantees for EVO.

### 4.1. Extreme Quantile Constraint

To quantify the probability of constraint violations, we employ a constrained quantile function that ensures the constraint is satisfied at a specified quantile level:

$$\arg \max_{\pi \in \Pi} J_R(\pi)$$
$$s.t. \quad q_\mu \leq d \tag{4}$$

where the quantile for the distribution of the cumulative cost $C = \sum_{t=0}^{\infty} \gamma^t c$ under policy $\pi$ is defined as:

$$q_\mu = \inf\{x|Pr[\sum_{t=0}^{\infty} \gamma^t c \leq x] \geq \mu\} \leq d \tag{5}$$

and $\mu$ is the quantile level. When the quantile satisfies the constraint threshold $q_\mu \leq d$, the probability $P(C \leq d) \geq \mu$ according to the definition of the quantile.

Extreme samples in the cost distribution tails are crucial for reducing the probability of constraint violations, as they carry critical signals strongly correlated with task constraints. To capture extreme samples, we propose an extreme quantile constraint that explicitly incorporates tail behavior:

$$q_{\mu+\nu} \leq d \qquad (6)$$

We denote $q_\mu$ in $q_{\mu+\nu}$ as the safety boundary, dividing the distribution into the body and the tail to capture extreme samples. $q_{\mu+\nu}$ denotes the risk boundary, integrating extreme values into the overall distribution. $\nu$ denotes the exploitation range in the tail, shown in Figure 2. The proposed quantile-based optimization objective minimizes constraint violations by optimizing the risk boundary, effectively accounting for extreme values in the distribution.

Extreme samples in the tail distribution are characterized by low probabilities and limited sample scale, posing challenges for accurate modeling. To address this, we leverage Extreme Value Theory to effectively capture extreme samples and reduce variance.

As stated in Theorem 3.1, samples exceeding a threshold follow a Generalized Pareto Distribution (GPD). To validate this, we empirically analyze cost samples from the environment, fitting the cumulative cost distribution to both a GPD and a Gaussian distribution. The results in Figure 1b demonstrate that the tail distribution better aligns with the GPD. By accurately capturing the tail behavior, the GPD enables more effective modeling of extreme samples, which are crucial for reducing constraint violations. Building on this insight, we propose an EVT-based approach for tail distribution modeling.

We first establish the relationship between EVT-based tail distribution modeling and the proposed quantile objective 6. Let $z = q_{\mu+\nu} - q_\mu > 0$ denote the excess value beyond the safety boundary $q_\mu$, used to explicitly model the tail. The cumulative distribution function (CDF) of $C$ is given by $F_C(q_{\mu+\nu}) = P(C \leq q_{\mu+\nu}) = P(C \leq q_\mu + z)$. We decompose the distribution of $C$ into two parts: the body distribution below $q_\mu$ and the tail distribution beyond $q_\mu$:

$$\begin{aligned} F_C(q_{\mu+\nu}) &= P(C \leq q_\mu + z) \\ &= P(C \leq q_\mu) + P(C > q_\mu)P(C - q_\mu \leq z | C > q_\mu) \end{aligned} \qquad (7)$$

Let $Z = C - q_\mu$ represent the excess random variable over $q_\mu$. We use the GPD to model the tail distribution $Z$, where $F_H(z)$ denotes the CDF of the GPD. According to Theorem 3.1, the conditional probability $P(C - q_\mu \leq z | C > q_\mu)$ in Equation 7 asymptotically follows the GPD $F_H(z)$. Thus $P(C \leq q_\mu + z)$ can be expressed as:

$$P(C \leq q_\mu + z) = \mu + \nu \simeq \mu + (1 - \mu)P(Z \leq z) \qquad (8)$$

Then we derive the quantile level in the GPD from Equation 8: $P(Z \leq z) \simeq \frac{\nu}{1-\mu}$. Denote the corresponding quantile in the GPD as $q^H$, then the risk boundary quantile $q_{\mu+\nu}$ is expressed using the quantile of GPD:

$$q_{\mu+\nu} = q_\mu + z \simeq q_\mu + q^H_{\frac{\nu}{1-\mu}} \qquad (9)$$

where the excess value $z$ is asymptotically equal to $q^H_{\frac{\nu}{1-\mu}}$ under the GPD. Furthermore, the relationship between the probability density function $f_C$ of $C$ and $f_H$ of GPD is:

$$f_C(q_\mu + z) = (1 - \mu)f_H(z) \qquad (10)$$

Then we propose a EVT-based optimization objective:

$$\begin{aligned} &\arg\max_{\pi \in \Pi} J_R(\pi) \\ &s.t. \quad q_\mu + q^H_{\frac{\nu}{1-\mu}} \leq d \end{aligned} \qquad (11)$$

where the constraint term denotes:

$$\begin{aligned} &\inf\{x | Pr[C \leq x] \geq \mu\} + \\ &\inf\{z | Pr[Z \leq z | z > 0] \geq \frac{\nu}{1-\mu}\} \leq d \end{aligned} \qquad (12)$$

In our work, the safety boundary $q_\mu$ is determined by the expectation of the cumulative cost, reflecting the average behavior of the cost distribution. In the trust region, we give the surrogate optimization objective:

$$\begin{aligned} &\pi_{k+1} = \arg\max_{\pi \in \Pi_\theta} \mathbb{E}_{s \sim d^{\pi_k}, a \sim \pi}[A^{\pi_k}_R(s, a)] \\ &J_C(\pi_k) + \frac{1}{1-\gamma}\mathbb{E}_{s \sim d^{\pi_k}, a \sim \pi}[A^{\pi_k}_C(s, a)] + q^H_{\frac{\nu}{1-\mu}} \leq d \\ &D(\pi \| \pi_k) \leq \delta \end{aligned} \qquad (13)$$

where $\Pi_\theta$ is the policy set parameterized by $\theta$, $D(\pi \| \pi_k) = \mathbb{E}_{s \sim d^{\pi_k}}[D_{KL}(\pi \| \pi_k)[s]]$, $D_{KL}$ is the KL divergence and $\delta > 0$ is the trust region size. $\{\pi \in \Pi_\theta : D(\pi \| \pi_k) \leq \delta\}$ defines the trust region. We provide the detailed optimization methods to solve the objective 13 in Appendix A.

As shown in Figure 2, during training, samples generated by the current policy $\pi$ are used to compute the safety boundary $q_\mu$. Samples where $C > q_\mu$, called peaks, are collected to form the peak set $Y_\mu$. The peaks are then used to fit the GPD, enabling the update of the risk boundary as $q_{\mu+\nu} = q_\mu + q^H_{\frac{\nu}{1-\mu}}$. Samples exceeding the updated risk boundary $q_\mu + q^H_{\frac{\nu}{1-\mu}}$ are identified as extreme samples and utilized for further analysis and policy optimization.

The parameters of the GPD, $\xi$ (shape) and $\sigma$ (scale), are estimated using maximum likelihood estimation:

$$\log \mathcal{L}(\xi, \sigma) = -N_\mu \log \sigma - \left(1 + \frac{1}{\xi}\right) \sum_{i=1}^{N_\mu} \log\left(1 + \frac{\xi}{\sigma} Y_i\right) \qquad (14)$$

where $Y_i$ denotes the sample in the peak set, $N_\mu$ is the number of peaks. The risk boundary can be obtained by:

$$q_\mu + q_{\frac{H_\nu}{1-\mu}} = q_\mu + \frac{\sigma}{\xi}\left((1-\frac{\nu n}{N_\mu})^{-\xi}-1\right) \quad (15)$$

where $n$ is the total number of samples.

## 4.2. Extreme Prioritization Resampling

Extreme samples in CRL are essential for guiding policy optimization. Extreme reward samples provide valuable insights for achieving task goals, while extreme cost samples offer critical information for satisfying task constraints. However, their low occurrence probability results in a rarity of these information-rich samples. To learn an optimal safe policy, it is essential to effectively exploit the scarce extreme samples. We model extreme reward and cost samples based on EVT and propose an extreme prioritization mechanism to enhance exploitation of learning signals within extreme samples during experience replay.

As discussed in Section 4.1, extreme cost samples are detected using EVT when their cost value exceeds the risk boundary. These samples form the extreme cost set, denoted as $Z_C : \{C > q_\mu + q_{\frac{H_\nu}{1-\mu}}\}$.

Similarly, EVT is applied to detect extreme reward samples in the replay buffer. The quantile $q_\mu^r$ is computed based on the expectation reward value $A_R$. Samples exceeding $q_\mu^r$, termed reward peaks, are collected into the reward peak set: $Y_\mu^r = \{A_R(s,a) > q_\mu^r\}$. The reward peak set is then used to fit by GPD, which determines the reward boundary $q_\mu^r + q_{\frac{H,r}{1-\mu}}$, following the process outlined in Equation 14 and 15. Samples with reward values exceeding the reward boundary have a high return to achieve the task goal, termed extreme reward samples. Denote the extreme reward set as $Z_R : \{A_R > q_\mu^r + q_{\frac{H,r}{1-\mu}}\}$.

To effectively exploit the learning signals in extreme samples, we assign higher priorities to these samples based on their quantile levels under the GPD. Samples with larger quantile levels typically indicate lower probabilities and higher value, and thus are given higher prioritization. The prioritization score $p$ is composed of two components: reward prioritization and cost prioritization. Reward prioritization is determined by the quantile level $\omega_r$ of a sample under the reward GPD, while cost prioritization is based on the quantile level $\omega_c$ under the cost GPD:

$$p = \omega_r + \omega_c \quad (16)$$

The probability of a sample $s_i$ being replayed with extreme prioritization is:

$$P(s_i) = \frac{p(s_i)}{\sum_{k=1}^{N} p(s_k)} \quad (17)$$

where $N$ is the total number of samples in the buffer. The extreme prioritization mechanism directly correlates a sample's replay frequency with its quantile levels in the reward and cost GPDs, emphasizing high-information extreme samples during experience replay.

## 4.3. Importance Resample for Extreme Samples

The GPD is fitted using a sparse extreme set in EVT. However, limited samples can lead to high variance, as individual extreme values exert a significant influence.

To address this issue, we introduce an off-policy importance resampling approach to augment extreme samples and reduce the variance of the GPD. By leveraging stored samples generated under previous policies $\pi_0$, we apply importance sampling to correct for the distributional shifts between samples generated by earlier policies and current policy $\pi$. Then the resampled values of previous samples $A_R$ and $C$ can be denoted as:

$$A_R' = \frac{\pi(a|s)}{\pi_0(a|s)}A_R, \quad C' = \frac{\pi(a|s)}{\pi_0(a|s)}C \quad (18)$$

By augmenting the scale of extreme samples through off-policy importance resampling, the variance of the GPD is effectively reduced, enhancing its stability, as supported by Theorem 4.3.

The algorithm process is presented in Algorithm 1.

## 4.4. Theoretical Analysis

We provide an upper bound on the expected constraint violation during policy updates in EVO.

**Theorem 4.1** (Constraint violation upper bound). *Suppose $\pi_{k+1}$, $\pi_k$ are related by quantile-based constraint objective 11, the upper bound on constraint of $\pi_{k+1}$ is:*

$$J_C(\pi_{k+1}) \leq d - q_{\frac{H_\nu}{1-\mu}}(\pi_{k+1})$$
$$+\frac{1}{1-\gamma}\mathbb{E}_{s\sim d^{\pi_k},a\sim\pi_{k+1}}\left[\frac{2\gamma\epsilon_C^{\pi_{k+1}}}{1-\gamma}D_{TV}(\pi_{k+1}\|\pi_k)[s]\right]$$
$$(19)$$

*where $\epsilon_C^{\pi_{k+1}} := \max_s |\mathbb{E}_{a\sim\pi_{k+1}}[A_C^\pi(s,a)]|$, TV-divergence $D_{TV}(\pi_{k+1}\|\pi_k)[s] = (1/2)\sum_a |\pi_{k+1}(a|s) - \pi_k(a|s)|$. The zero-violation exploitation range $\nu_0$ in GPD satisfies:*

$$\nu_0 = \frac{N_\mu}{n}\left(1 - \left(\frac{\xi}{\sigma(1-\gamma)}\right.\right.$$
$$\left.\left.\mathbb{E}_{s\sim d^{\pi_k},a\sim\pi_{k+1}}\left[\frac{2\gamma\epsilon_C^{\pi_{k+1}}}{1-\gamma}D_{TV}(\pi_{k+1}\|\pi_k)[s]\right]+1\right)^{-\frac{1}{\xi}}\right)$$
$$(20)$$

*which guarantees the expectation of updated policy $\pi_{k+1}$ strictly satisfies the constraints, where $N_\mu$ is the number of peaks and $n$ is the total number of samples.*

---

**Algorithm 1** EVO: Extreme Value policy Optimization

---

**Input:** Initialize policy network $\pi_\theta$, value networks $V_R^\psi$, $V_C^\upsilon$.

**Output:** The optimal policy parameter $\theta$.

1: **for** epoch $k = 0, 1, 2, ...$ **do**
2:     Sample trajectories under the current policy $\pi_{\theta_k}$.
3:     Process the trajectories to $C$-returns, and calculate advantage functions with $V_R^\psi$ and $V_C^\upsilon$.
4:     **for** $K$ iterations **do**
5:         Update value networks $V_R^\psi$ and $V_C^\upsilon$.
6:     **end for**
7:     Perform importance resampling to adjust off-policy samples using Eq. 18.
8:     Fit the cost GPD with off-policy samples according to Eq. 14.
9:     Compute the risk boundary $q_\mu + q_{\frac{H_\nu}{1-\mu}}$ using the fitted cost GPD (Eq. 11).
10:    Fit the reward GPD with off-policy samples according to Eq. 14.
11:    Compute the extreme prioritization (Eq. 17) from the reward and cost GPDs.
12:    Update policy by optimizing the new objective (Eq. 11) using on-policy samples.
13: **end for**
14: **Return:** Policy parameters $\theta = \theta_{k+1}$.

---

A proof is provided in Appendix B.1.

Theorem 4.1 demonstrates that EVO has a tighter constraint violation upper bound compared to the expectation optimization objective in CPO (Achiam et al., 2017). Theoretically, with the zero-violation exploitation range $\nu_0$, EVO guarantees that the expectation of the updated policy $\pi_{k+1}$ strictly satisfies the constraints.

Additionally, we analyze the probability of constraint violations under the quantile-based objective in EVO.

**Theorem 4.2** (Constraint violation probability)**.** *When the safety boundary $q_\mu$ is determined by the expectation of cumulative cost $C$, the constraint violation probability of EVO with the zero-violation exploitation range $\nu_0$ satisfies:*

$$P(C > d) < (\mathcal{J}(\mathcal{E}+1))^{-\frac{1}{\xi}} \tag{21}$$

*where:*

$$\mathcal{J} = \frac{\xi}{\sigma}\left(J_C(\pi_k) + \frac{1}{1-\gamma}\mathbb{E}_{s\sim d^{\pi_k}, a\sim\pi}[A_C^{\pi_k}(s,a)]\right) + 1$$

$$\mathcal{E} = \frac{\xi}{\sigma(1-\gamma)}\mathbb{E}_{s\sim d^{\pi_k}, a\sim\pi_{k+1}}\left[\frac{2\gamma\epsilon_C^{\pi_{k+1}}}{1-\gamma}D_{TV}(\pi_{k+1}\|\pi_k)[s]\right] \tag{22}$$

*EVO has a lower constraint violation probability than the expectation-based CRL by a margin of $\nu_0$.*

A proof is provided in Appendix B.2.

Theorem 4.2 indicates that EVO has a smaller constraint violation probability than the expectation-based objective when $\mu$ corresponds to the quantile level of the constraint expectation.

**Theorem 4.3** (Variance of EVO)**.** *Let $q_{\frac{H_\nu}{1-\mu}}$ denote the quantile in GPD for random samples of size $N$ from a population with inverse distribution function $q_{\frac{H_\nu}{1-\mu}} = F_H^{-1}(\frac{\nu}{1-\mu})$. If the cumulative distribution function $F_H$ is continuous and has continuous and positive density $f_H$ at $q_{\frac{H_\nu}{1-\mu}}$, then the quantile estimator $q_{\frac{H_\nu}{1-\mu}}$ to the true quantile value $q^*_{\frac{\nu}{1-\mu}}$ converges in distribution to an Gaussian random vector with mean 0 and variance $\Omega$:*

$$\sqrt{N}(q_{\frac{H_\nu}{1-\mu}} - q^*_{\frac{\nu}{1-\mu}}) \to \mathcal{N}(0, \Omega) \tag{23}$$

*The variance of EVO to estimate the quantile $\frac{\nu}{1-\mu}$ is:*

$$\Omega = \frac{\nu(1-\mu-\nu)}{N(1-\mu)^2 f_H^2(q_{\frac{H_\nu}{1-\mu}})} \tag{24}$$

*where $\mu$ denotes the safety boundary quantile, $\nu$ denotes the exploitation range, and $N$ denotes the number of samples from the distribution that are used to estimate the quantile values. EVO has a lower variance than that in quantile regression methods:*

$$\Omega_2 = \frac{(\mu+\nu)(1-\mu-\nu)}{N f_C^2(q_{\mu+\nu})} \tag{25}$$

*where $f_C$ denotes the probability density function.*

A proof is provided in Appendix B.3.

Note that Theorem 4.3 indicates the variance of EVO is negatively correlated with the number of samples used to estimate the quantile, which demonstrates that extending the extreme sample scale through off-policy importance resampling effectively reduces the variance of EVO.

## 5. Experiment

We evaluate the performance of EVO through comparison with multiple baselines, including the expectation-based constraint method CPO (Achiam et al., 2017), the probability-based constraint method WCSAC (Yang et al., 2021), and state augmentation approaches Saute (Sootla et al., 2022a) and Simmer (Sootla et al., 2022b), which emphasize zero-constraint violation. Results for additional baselines are provided in Appendix C.1.1. The experiments address the following questions: 1) Does EVO reduce constraint violations while maintaining policy performance compared to baselines? 2) Can the modeling and exploitation of extreme samples within EVO help mitigate variance and enhance policy learning?

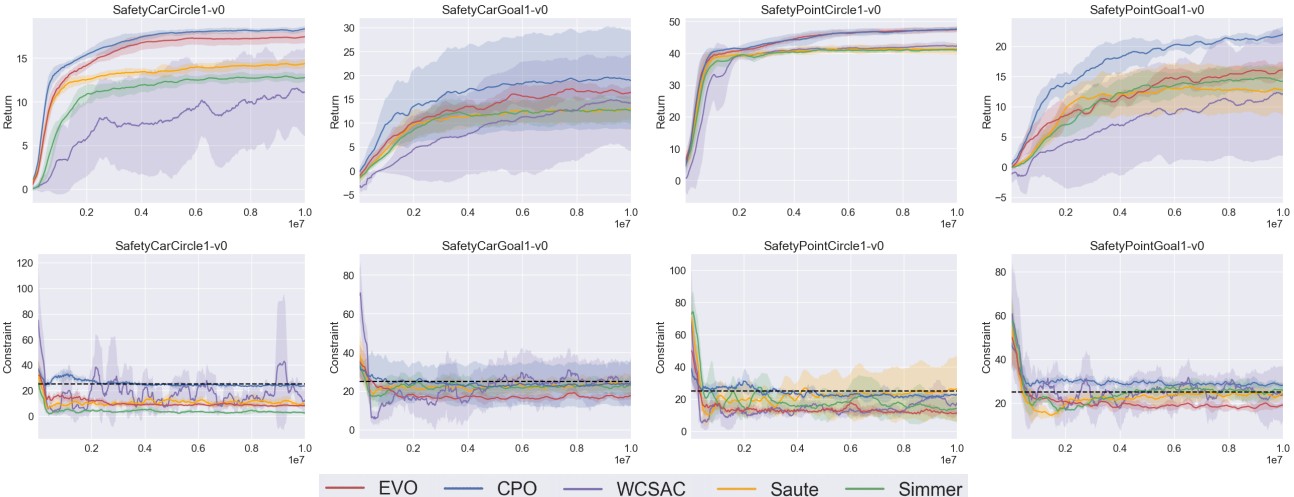

Figure 3: Comparison of EVO to baselines on Safety Gym. The x-axis is the total number of training steps, the y-axis is the average return or constraint. The solid line is the mean and the shaded area is the standard deviation. The dashed line is the constraint threshold which is 25.

**Environment.** The environments in our experiments consist of Safety Gymnasium and MuJoCo (Ji et al., 2024). The Safety Gymnasium tasks simulate a simplified version of autonomous driving, where the robot is required to reach a goal while avoiding obstacles. The Safety MuJoCo tasks focus on robot motion control, with agents being rewarded for maintaining a straight path while adhering to a speed limit to ensure safety and stability. Details are provided in Appendix C.3.

**Experiment Setting.** All experiments followed uniform conditions to ensure fairness and reproducibility, with a total of $10^7$ training time steps and a maximum trajectory length of 1000 steps. To reduce randomness, 6 random seeds were used for each method, and the results are presented as mean and variance. The parameter settings are in Appendix C.4.

**Performance and Constraint.** In the navigation tasks of Safety Gymnasium, the initial policy is infeasible, presenting significant challenges due to multiple obstacles and complex decision boundaries. Figure 3 shows the learning curves of EVO and baselines. The results indicate that EVO rapidly converges to the feasible domain and subsequently maintains zero constraint violations throughout training. This is attributed to its optimization objective, which leverages extreme constraints to effectively capture violation signals from extreme cost values, thereby significantly reducing the probability of constraint violations, as supported by Theorems 4.1 and 4.2. Furthermore, EVO achieves policy performance comparable to CPO while outperforming other methods such as Saute and WCSAC that emphasize zero constraint violations. It is important to note that CPO violates the constraint threshold in SafetyPointGoal1-v0, so a higher return than EVO does not indicate a better policy. In

contrast, EVO achieves the best overall performance while strictly adhering to the constraint threshold. In the Safety MuJoCo tasks, as illustrated in Figure 4, when the initial policy lies within the feasible domain, EVO maintains zero constraint violations throughout training, providing strong empirical validation of the zero-violation guarantee in Theorem 4.1. Despite using a probabilistic constraint optimization objective, WCSAC presents significant constraint violations in SafetyCarCircle1-v0 and SafetyHalfCheetahVelocity-v1, because it fails to account for the substantial impact of extreme samples. We design an evaluation metric to comprehensively measure policy performance and constraint satisfaction during training, results in Appendix C.2 show that EVO outperforms across multiple environments.

**Ablation Study.** Ablation experiments are designed to validate the effectiveness of each component in EVO. (1) Ablating EVT-based Constraint Objective: We replace the EVT-based constraint objective with a constant quantile constraint objective. The results in Figure 5a and 5b show that the constant quantile objective achieves constraint satisfaction by diminishing policy performance. Conversely, the EVT-based constraint objective in EVO, derived from the extreme samples in the tail, captures critical information about the task goal. This enables EVO to achieve a more favorable trade-off between policy performance and constraint satisfaction. (2) Ablating Extreme Prioritization: The results in Figure 5a and 5b show a degradation in performance when extreme prioritization during experience replay is removed. This indicates that prioritizing extreme samples enhances the optimization process by fully exploiting the task-related learning signals embedded in these low-probability, high-impact samples. (3) Ablating Off-policy Importance Resampling for GPD Fitting: As shown in Figure 6, eliminating

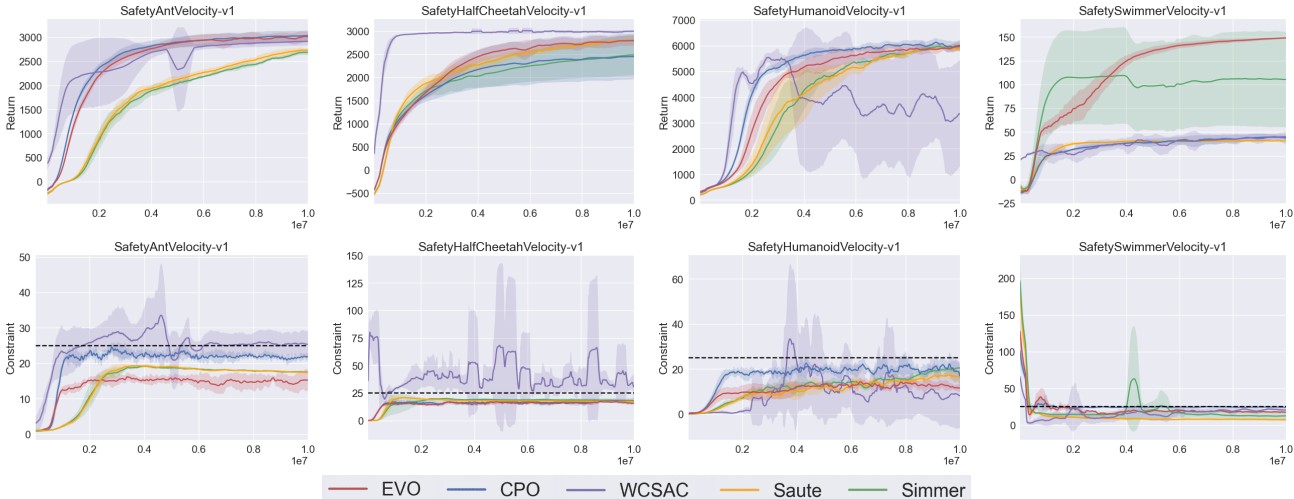

Figure 4: Comparison of EVO to baselines on Safety MuJoCo. The x-axis is the total number of training steps, the y-axis is the average return or constraint. The solid line is the mean and the shaded area is the standard deviation. The dashed line is the constraint threshold which is 25.

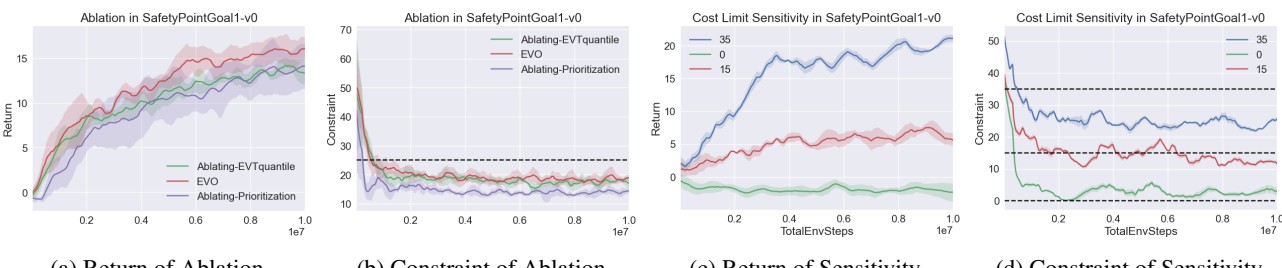

(a) Return of Ablation     (b) Constraint of Ablation     (c) Return of Sensitivity     (d) Constraint of Sensitivity

Figure 5: (a)(b) Ablation Study, including ablating EVT-based Constraint Objectives and ablating extreme prioritization. (c)(d) Cost Limit Sensitivity.

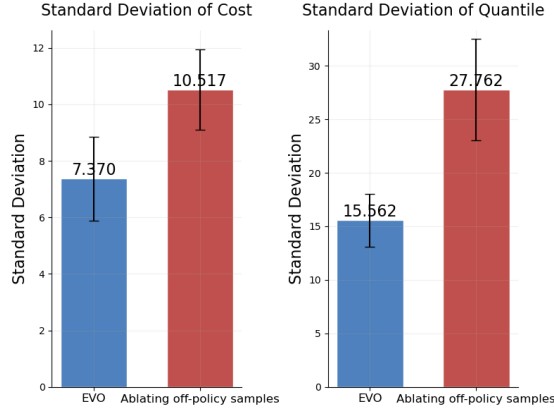

Figure 6: Ablating off-policy samples for GPD fitting.

off-policy samples leads to an increase in the variance of the GPD, providing empirical validation of Theorem 4.3. This effect is due to the sparsity of extreme samples, particularly cost extremes, which become increasingly rare as training progresses. Without sample augmentation, GPD fitting becomes overly sensitive to individual extreme values, leading

to inaccurate tail modeling.

**Cost Limit Sensitivity.** We further evaluate the robustness of EVO across various threshold levels. Figure 5c and 5d indicate that EVO effectively satisfies different cost limits while maintaining policy performance. The ability to learn from extreme values enables EVO to adapt to diverse policy preferences. In security-oriented settings with limit 0, cost extremes in EVO provide learning signals related to task constraints, and in performance-oriented settings with limit 35, reward extremes in EVO offer significant reward signals about the task goals.

**The Fitting Accuracy of GPD.** To evaluate the fitting accuracy of GPD in our experiments, we collected training data across multiple environments and fitted both GPD and Gaussian distributions. Furthermore, we employed the Kolmogorov-Smirnov (KS) test to quantify the fit accuracy, where lower values indicate more accurate fits. As shown in Figure 7, the GPD presents robust fitting performance across various data distributions. It provides a more accurate fit for extreme samples than the Gaussian and more effectively

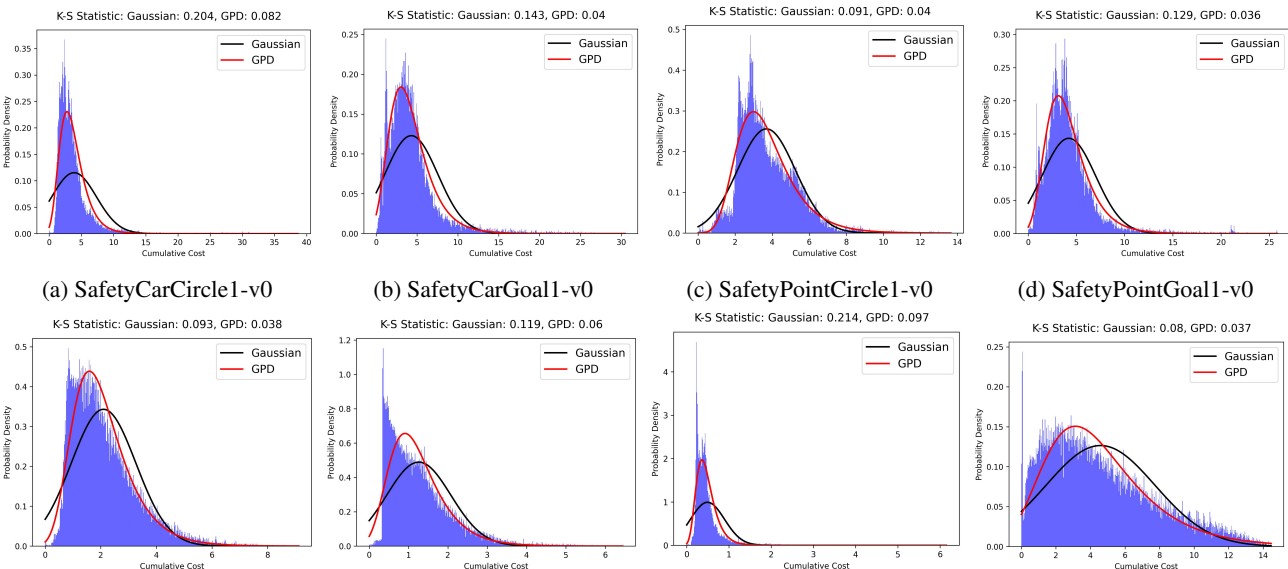

(a) SafetyCarCircle1-v0     (b) SafetyCarGoal1-v0     (c) SafetyPointCircle1-v0     (d) SafetyPointGoal1-v0

(e) SafetyAntVelocity-v1    (f) SafetyHalfCheetahVelocity-v1   (g) SafetyHumanoidVelocity-v1   (h) SafetySwimmerVelocity-v1

Figure 7: Distribution curves fitted by GPD and Gaussian on the training data in EVO. The fitting accuracy is quantified using the Kolmogorov-Smirnov test, as shown in the titles, where lower values indicate higher fitting accuracy. GPD presents accurate fitting performance across various data distributions.

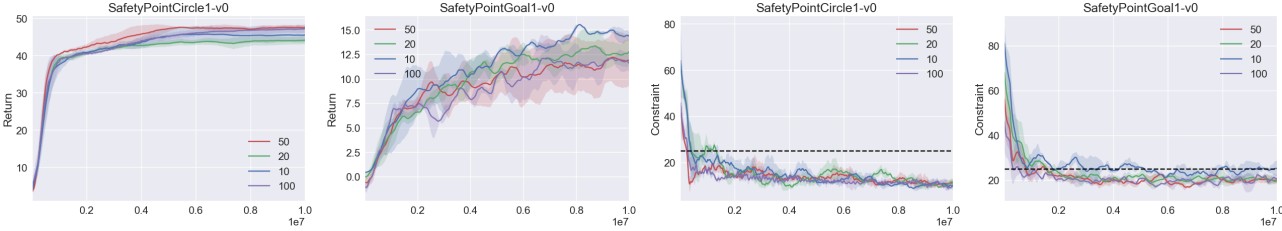

Figure 8: The training curves of EVO with different sample sizes across multiple environments, indicating that EVO remains effective even with small sample sets.

capturing the tail behavior of the distribution. However, in special cases where the gap between extreme and normal values is subtle, the GPD may not provide a satisfactory fit due to the requirement in EVT that the threshold $t$ be sufficiently large. To address this issue, one can employ non-linear transformations to amplify the differences between extreme and normal values prior to GPD fitting, thereby enhancing the accuracy of tail modeling.

**Sample Size of GPD Fitting.** We conducted experiments by varying the sample size used for GPD fitting and evaluated the corresponding policy performance. As shown in Figure 8, increasing the sample size generally lead to improved constraint satisfaction. Notably, in SafetyPointCircle1-v0, EVO maintains strong constraint satisfaction and performance even with as few as 10 samples. In SafetyPointGoal1-v0, constraint satisfaction is consistently achieved once the sample size exceeds 20. In our experiments, with the same sample size of 20, EVO demonstrates superior constraint satisfaction and policy perfor-

mance compared to baselines, indicating that it remains effective even with small sample sets.

## 6. Conclusion

In this paper, we propose a zero-constraint violation method, Extreme Value policy Optimization (EVO) algorithm. EVO leverages Extreme Value Theory (EVT) to model extreme samples and exploits them to enhance safety policy optimization. To address the challenge of high variance in extreme samples, we employ the GPD to explicitly model the tail extremes and augment the extreme samples with off-policy samples. Recognizing the high information of extreme samples, we introduce an extreme prioritization mechanism to exploit them. We provide both theoretical and experimental evidence that EVO achieves zero constraint violations during training, while maintaining high returns and low variance.

## Acknowledgment

This work was supported by NSF China under Grant No. T2421002, 62202299, 62020106005, 62061146002.

## Impact Statement

This paper aims to advance the field of Reinforcement Learning by introducing the EVO algorithm, which significantly improves safety of policy during RL training. EVO enables safer and more efficient deployment of RL in real-world domains such as large language models, autonomous driving, and robotics, thereby contributing to the broader progress of Machine Learning. There are potential societal consequences of our work, none which we feel must be specifically highlighted here.

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

**Notations**

| | |
|---|---|
| $c$ | cost |
| $C$ | cumulative cost |
| $d$ | constraint threshold |
| $A_R$ | reward advantage function |
| $A_C$ | cost advantage function |
| $q$ | quantile |
| $\mu$ | quantile level |
| $q_\mu$ | safety boundary |
| $\nu$ | exploitation range in extrem samples |
| $q_{\mu+\nu}$ | risk boundary |
| $z$ | excess value beyond the safety boundary |
| $Z$ | excess random variable |
| $F$ | cumulative distribution function |
| $f$ | probability density function |
| $q^H$ | quantile in GPD |
| $\xi$ | shape parameter of GPD |
| $\sigma$ | scale parameter of GPD |
| $Y_\mu$ | peaks set |
| $Z_C$ | extreme cost set |
| $\delta$ | trust region size |

## A. Policy Optimization in EVO

The CRL aims to find an optimal policy by maximizing the expected discount return over the set of feasible policies $\Pi_C := \{\pi \in \Pi : J_C(\pi) \le d\}$:

$$\arg \max_{\pi \in \Pi} J_R(\pi)$$
$$s.t. \quad J_C(\pi) \le d \tag{26}$$

The following equation briefly gives the performance difference of arbitrary two policies (Schulman et al., 2015a), which represents the expected return of another policy $\pi'$ in terms of the advantage function over $\pi$:

$$J_R(\pi') - J_R(\pi) = \frac{1}{1-\gamma} \mathbb{E}_{s \sim d^{\pi'}, a \sim \pi'}[A_R^\pi(s,a)] \tag{27}$$

This implies that iterative updates to the policy, $\pi'(s) = \arg \max_a A_R^\pi(s,a)$, lead to performance improvement until convergence to the optimal solution.

According to the performance difference equation (27), CRL is defined as a constrained optimization problem:

$$\pi_{k+1} = \arg\max_{\pi \in \Pi_\theta} \mathbb{E}_{s \sim d^\pi, a \sim \pi}[A_R^{\pi_k}(s, a)]$$
$$s.t. \quad J_C(\pi_k) + \frac{1}{1-\gamma}\mathbb{E}_{s \sim d^\pi, a \sim \pi}[A_C^{\pi_k}(s, a)] \leq d \tag{28}$$

where policy $\pi \in \Pi_\theta$ is parameterized with parameters $\theta$, and $\pi_k$ represents the current policy.

In this paper, we propose the quantile-based objective,

$$\arg\max_{\pi \in \Pi} J_R(\pi)$$
$$s.t. \quad q_\mu + q_{\frac{\nu}{1-\mu}}^H \leq d \tag{29}$$

where the safety boundary quantile $q_\mu$ is determined by the expectation $J_C(\pi')$. Then the objective can be denoted as:

$$\pi_{k+1} = \arg\max_{\pi \in \Pi_\theta} \mathbb{E}_{s \sim d^\pi, a \sim \pi}[A_R^{\pi_k}(s, a)]$$
$$s.t. \quad J_C(\pi_k) + \frac{1}{1-\gamma}\mathbb{E}_{s \sim d^\pi, a \sim \pi}[A_C^{\pi_k}(s, a)] + q_{\frac{\nu}{1-\mu}}^H \leq d \tag{30}$$

We propose the optimization method based on CPO, EVO-CPO, to solve the optimization 29.

## A.1. CPO-based EVO

The complex dependency of state visitation distribution $d^\pi(s)$ on unknown policy $\pi$ makes Equation 29 difficult to optimize directly. To address this, this paper uses samples generated by the current policy $\pi_k$ to approximate the original problem locally. We seek to solve the following optimization problem in the trust region:

$$\pi_{k+1} = \arg\max_{\pi \in \Pi_\theta} \mathbb{E}_{s \sim d^{\pi_k}, a \sim \pi}[A_R^{\pi_k}(s, a)]$$
$$s.t. \quad J_C(\pi_k) + \frac{1}{1-\gamma}\mathbb{E}_{s \sim d^{\pi_k}, a \sim \pi}[A_C^{EI}(s, a|\pi_k)] + q_{\frac{\nu}{1-\mu}}^H \leq d \tag{31}$$
$$D(\pi\|\pi_k) \leq \delta$$

where $\Pi_\theta$ is the policy set parameterized by parameter $\theta$, $D(\pi\|\pi_k) = \mathbb{E}_{s \sim d^{\pi_k}}[D_{KL}(\pi\|\pi_k)[s]]$, $D_{KL}$ is the KL divergence and $\delta > 0$ is the step size. The set $\{\pi \in \Pi_\theta : D(\pi\|\pi_k) \leq \delta\}$ is the trust region.

In the EVO-CPO method, we approximate the reward objective and cost constraints with first-order expansion and approximate the KL-divergence constraint with second-order expansion. The local approximation to Equation 31 is:

$$\theta_{k+1} = \arg\max_\theta g^T(\theta - \theta_k)$$
$$s.t. \quad c + (g_C)^T(\theta - \theta_k) \leq 0 \tag{32}$$
$$\frac{1}{2}(\theta - \theta_k)^T H(\theta - \theta_k) \leq \delta$$

where $g$ denotes the gradient of the reward objective in equation 31, $g_C$ denotes the gradient of quantile-based constraint in Equation 31, $c = J_C(\pi_k) - d$, $H$ is the Hessian of the KL-divergence. When the constraint is satisfied, the analytical solution can be obtained using the primal-dual method. The solution to the primal problem is:

$$\theta^* = \theta_k + \frac{1}{\lambda^*}H^{-1}(g - g_C\nu^*) \tag{33}$$

where $\lambda$ and $\nu$ are the Lagrangian multipliers of the KL-divergence term and the constraint term in the Lagrangian function, respectively. $\lambda^*$, $\nu^*$ are the solutions to the dual problem:

$$\nu^* = \max\{0, \frac{\lambda^* c - u}{v}\} \tag{34}$$

$$\lambda^* = \arg\max_{\lambda \geq 0} \begin{cases} \frac{1}{2\lambda}\left(\frac{u^2}{v} - q\right) + \frac{\lambda}{2}\left(\frac{c^2}{v} - \delta\right) - \frac{uc}{v}, & \text{if}\lambda c > u \\ -\frac{1}{2}\left(\frac{q}{\lambda} + \lambda\delta\right), \text{otherwise,} \end{cases} \tag{35}$$

where $q = g^T H^{-1} g$, $u = g^T H^{-1} g_C$, $v = (g_C)^T H^{-1} g_C$.

When the constraint is violated, we use the conjugate gradient method to decrease the constraint value:

$$\theta^* = \theta_k - \left(\frac{2\delta}{(g_C)^T H^{-1} g_C}\right)^{\frac{1}{2}} H^{-1} g_C \tag{36}$$

## B. Theoretical Proof

### B.1. Constraint Violation Upper Bound

**Lemma B.1.** *(Achiam et al., 2017) For any policies $\pi'$, $\pi$, and any cost function $C : S \to \mathbb{R}$, with $\epsilon_C^{\pi'} = \max_s |\mathbb{E}_{a\sim\pi'}[A_C^\pi(s,a)]|$. The following bound holds:*

$$J_C(\pi') - J_C(\pi) \leq \frac{1}{1-\gamma}\mathbb{E}_{s\sim d^\pi, a\sim\pi'}[A_C^\pi(s,a) + \frac{2\gamma\epsilon_C^{\pi'}}{1-\gamma}D_{TV}(\pi'\|\pi)[s]] \tag{37}$$

*where $D_{TV}(\pi'\|\pi)[s] = (1/2)\sum_a |\pi'(a|s) - \pi(a|s)|$ is the total variational divergence between action distribution at $s$.*

**Theorem B.2** (Constraint violation upper bound). *Suppose $\pi_{k+1}$, $\pi_k$ are related by quantile-based constraint objective 11, the upper bound on constraint of $\pi_{k+1}$ is:*

$$J_C(\pi_{k+1}) \leq d - q_{\frac{H_\nu}{1-\mu}}(\pi_{k+1}) + \frac{1}{1-\gamma}\mathbb{E}_{s\sim d^{\pi_k}, a\sim\pi_{k+1}}[\frac{2\gamma\epsilon_C^{\pi_{k+1}}}{1-\gamma}D_{TV}(\pi_{k+1}\|\pi_k)[s]] \tag{38}$$

*where $\epsilon_C^{\pi_{k+1}} := \max_s |\mathbb{E}_{a\sim\pi_{k+1}}[A_C^\pi(s,a)]|$, $D_{TV}(\pi_{k+1}\|\pi_k)[s] = (1/2)\sum_a |\pi_{k+1}(a|s) - \pi_k(a|s)|$. The zero-violation exploitation range $\nu_0$ in GPD satisfies:*

$$\nu_0 = \frac{N_\mu}{n}\left(1 - \left(\frac{\xi}{\sigma(1-\gamma)}\mathbb{E}_{s\sim d^{\pi_k}, a\sim\pi_{k+1}}[\frac{2\gamma\epsilon_C^{\pi_{k+1}}}{1-\gamma}D_{TV}(\pi_{k+1}\|\pi_k)[s]] + 1\right)^{-\frac{1}{\xi}}\right) \tag{39}$$

*which guarantees the expectation of updated policy $\pi_{k+1}$ strictly satisfies the constraints, where $N_\mu$ is the number of peaks and $n$ is the total number of samples.*

*Proof.* According to the Lemma B.1 in CPO (Achiam et al., 2017), for any policies $\pi'$ and $\pi$, and any cost function, the following bound holds:

$$J_C(\pi') - J_C(\pi) \leq \frac{1}{1-\gamma}\mathbb{E}_{s\sim d^\pi, a\sim\pi'}[A_C^\pi(s,a) + \frac{2\gamma\epsilon_C^{\pi'}}{1-\gamma}D_{TV}(\pi'\|\pi)[s]] \tag{40}$$

Then the bounds of two neighboring policies $\pi_k$ and $\pi_{k+1}$ in the policy update follows:

$$J_C(\pi_{k+1}) - J_C(\pi_k) \leq \frac{1}{1-\gamma}\mathbb{E}_{s\sim d^{\pi_k}, a\sim\pi_{k+1}}[A_C^{\pi_k}(s,a) + \frac{2\gamma\epsilon_C^{\pi_{k+1}}}{1-\gamma}D_{TV}(\pi_{k+1}\|\pi_k)[s]] \tag{41}$$

$$J_C(\pi_{k+1}) \leq J_C(\pi_k) + \frac{1}{1-\gamma}\mathbb{E}_{s\sim d^{\pi_k}, a\sim\pi_{k+1}}[A_C^{\pi_k}(s,a) + \frac{2\gamma\epsilon_C^{\pi_{k+1}}}{1-\gamma}D_{TV}(\pi_{k+1}\|\pi_k)[s]] \tag{42}$$

Suppose $\pi_k$ and $\pi_{k+1}$ are related by the optimization objective in EVO, which is:

$$q_\mu + q_{\frac{H_\nu}{1-\mu}} \leq d \tag{43}$$

where we use the expectation $J_C(\pi)$ to denote the quantile value $q_\mu$, which is approximated in the trust region as:

$$J_C(\pi_k) + \frac{1}{1-\gamma}\mathbb{E}_{s\sim d^{\pi_k},a\sim\pi}[A_C^{\pi_k}(s,a)] + q_{\frac{H_\nu}{1-\mu}} \leq d \tag{44}$$

According to Equation 42 and Equation 44, then the upper bound on the constraint expectation of $\pi_{k+1}$ is:

$$
\begin{aligned}
J_C(\pi_{k+1}) \leq &J_C(\pi_k) + \frac{1}{1-\gamma}\mathbb{E}_{s\sim d^{\pi_k},a\sim\pi_{k+1}}[A_C^{\pi_k}(s,a)] + \frac{1}{1-\gamma}\mathbb{E}_{s\sim d^{\pi_k},a\sim\pi_{k+1}}[\frac{2\gamma\epsilon_C^{\pi_{k+1}}}{1-\gamma}D_{TV}(\pi_{k+1}\|\pi_k)[s]] \\
\leq &d - q_{\frac{H_\nu}{1-\mu}} + \frac{1}{1-\gamma}\mathbb{E}_{s\sim d^{\pi_k},a\sim\pi_{k+1}}[\frac{2\gamma\epsilon_C^{\pi_{k+1}}}{1-\gamma}D_{TV}(\pi_{k+1}\|\pi_k)[s]]
\end{aligned}
\tag{45}
$$

The risk boundary in GPD $q_{\frac{H_\nu}{1-\mu}} \geq 0$, which indicates that EVO has a tighter constraint violation upper bound compared to the expectation optimization objective in CPO, which is:

$$J_C(\pi_{k+1}) \leq d + \frac{1}{1-\gamma}\mathbb{E}_{s\sim d^{\pi_k},a\sim\pi_{k+1}}[\frac{2\gamma\epsilon_C^{\pi_{k+1}}}{1-\gamma}D_{TV}(\pi_{k+1}\|\pi_k)[s]] \tag{46}$$

Theoretically, if the value of $q_{\frac{H_\nu}{1-\mu}}$ is greater than $\frac{1}{1-\gamma}\mathbb{E}_{s\sim d^{\pi_k},a\sim\pi_{k+1}}[\frac{2\gamma\epsilon_C^{\pi_{k+1}}}{1-\gamma}D_{TV}(\pi_{k+1}\|\pi_k)[s]]$, our method ensures that the expectation of updated policy $\pi_{k+1}$ strictly satisfies the constraints.

The risk boundary in EVO is obtained by:

$$q_{\frac{H_\nu}{1-\mu}} = \frac{\sigma}{\xi}\left((1-\frac{\nu n}{N_\mu})^{-\xi} - 1\right) \tag{47}$$

We derive the zero-violation exploitation range $\nu_0$ by making:

$$
\begin{aligned}
q_{\frac{H_\nu}{1-\mu}} &= \frac{1}{1-\gamma}\mathbb{E}_{s\sim d^{\pi_k},a\sim\pi_{k+1}}[\frac{2\gamma\epsilon_C^{\pi_{k+1}}}{1-\gamma}D_{TV}(\pi_{k+1}\|\pi_k)[s]] \\
\frac{\sigma}{\xi}\left((1-\frac{\nu n}{N_\mu})^{-\xi} - 1\right) &= \frac{1}{1-\gamma}\mathbb{E}_{s\sim d^{\pi_k},a\sim\pi_{k+1}}[\frac{2\gamma\epsilon_C^{\pi_{k+1}}}{1-\gamma}D_{TV}(\pi_{k+1}\|\pi_k)[s]]
\end{aligned}
\tag{48}
$$

Then we get the zero-violation exploitation range $\nu_0$:

$$\nu_0 = \frac{N_\mu}{n}\left(1 - \left(\frac{\xi}{\sigma(1-\gamma)}\mathbb{E}_{s\sim d^{\pi_k},a\sim\pi_{k+1}}[\frac{2\gamma\epsilon_C^{\pi_{k+1}}}{1-\gamma}D_{TV}(\pi_{k+1}\|\pi_k)[s]] + 1\right)^{-\frac{1}{\xi}}\right) \tag{49}$$

$\square$

## B.2. Constraint Violation Probability

**Theorem B.3** (Constraint violation probability). *When the safety boundary $q_\mu$ is determined by the expectation of cumulative cost, the constraint violation probability of EVO with the zero-violation exploitation range $\nu_0$ satisfies:*

$$P(C > d) < (\mathcal{J}(\mathcal{E}+1))^{-\frac{1}{\xi}} \tag{50}$$

*where:*

$$
\begin{aligned}
\mathcal{J} &= \frac{\xi}{\sigma}\left(J_C(\pi_k) + \frac{1}{1-\gamma}\mathbb{E}_{s\sim d^{\pi_k},a\sim\pi}[A_C^{\pi_k}(s,a)]\right) + 1 \\
\mathcal{E} &= \frac{\xi}{\sigma(1-\gamma)}\mathbb{E}_{s\sim d^{\pi_k},a\sim\pi_{k+1}}[\frac{2\gamma\epsilon_C^{\pi_{k+1}}}{1-\gamma}D_{TV}(\pi_{k+1}\|\pi_k)[s]]
\end{aligned}
\tag{51}
$$

*EVO has a lower constraint violation probability than the expectation-based CRL by a margin of $\nu_0$.*

*Proof.* In this paper, we use the expectation $J_C(\pi)$ of cumulative cost $C$ to determine the safety boundary $q_\mu$:

$$q_\mu = J_C(\pi) \tag{52}$$

which is used to fit the GPD with:

$$1 - \left(1 + \frac{\xi x}{\sigma}\right)^{-\frac{1}{\xi}}, \quad \xi \neq 0 \tag{53}$$

Then we get the safety boundary quantile level $\mu$:

$$\mu = 1 - \left(1 + \frac{\xi J_C(\pi)}{\sigma}\right)^{-\frac{1}{\xi}} \tag{54}$$

where $J_C(\pi)$ is approximated in the local region as:

$$J_C(\pi) = J_C(\pi_k) + \frac{1}{1-\gamma}\mathbb{E}_{s \sim d^{\pi_k}, a \sim \pi}[A_C^{\pi_k}(s,a)] \tag{55}$$

Then we get the zero-violation exploration range $\nu_0$ according to Theorem B.2:

$$
\begin{aligned}
\nu_0 &= (1-\mu)\left(1 - \left(\frac{\xi}{\sigma(1-\gamma)}\mathbb{E}_{s \sim d^{\pi_k}, a \sim \pi_{k+1}}\left[\frac{2\gamma \epsilon_C^{\pi_{k+1}}}{1-\gamma}D_{TV}(\pi_{k+1}\|\pi_k)[s]] + 1\right)^{-\frac{1}{\xi}}\right) \\
&= \left(1 + \frac{\xi J_C(\pi)}{\sigma}\right)^{-\frac{1}{\xi}}\left(1 - \left(\frac{\xi}{\sigma(1-\gamma)}\mathbb{E}_{s \sim d^{\pi_k}, a \sim \pi_{k+1}}\left[\frac{2\gamma \epsilon_C^{\pi_{k+1}}}{1-\gamma}D_{TV}(\pi_{k+1}\|\pi_k)[s]] + 1\right)^{-\frac{1}{\xi}}\right)
\end{aligned}
\tag{56}
$$

According to the definition fo the quantile in EVO, the constraint violation probability of EVO satisfies:

$$
\begin{aligned}
&P(C > d) \\
&< 1 - \mu - \nu_0 \\
&= 1 - \mu - (1-\mu)\left(1 - \left(\frac{\xi}{\sigma(1-\gamma)}\mathbb{E}_{s \sim d^{\pi_k}, a \sim \pi_{k+1}}\left[\frac{2\gamma \epsilon_C^{\pi_{k+1}}}{1-\gamma}D_{TV}(\pi_{k+1}\|\pi_k)[s]] + 1\right)^{-\frac{1}{\xi}}\right) \\
&= (1-\mu)\left(\frac{\xi}{\sigma(1-\gamma)}\mathbb{E}_{s \sim d^{\pi_k}, a \sim \pi_{k+1}}\left[\frac{2\gamma \epsilon_C^{\pi_{k+1}}}{1-\gamma}D_{TV}(\pi_{k+1}\|\pi_k)[s]] + 1\right)^{-\frac{1}{\xi}} \\
&= \left(1 + \frac{\xi J_C(\pi)}{\sigma}\right)^{-\frac{1}{\xi}}\left(\frac{\xi}{\sigma(1-\gamma)}\mathbb{E}_{s \sim d^{\pi_k}, a \sim \pi_{k+1}}\left[\frac{2\gamma \epsilon_C^{\pi_{k+1}}}{1-\gamma}D_{TV}(\pi_{k+1}\|\pi_k)[s]] + 1\right)^{-\frac{1}{\xi}} \\
&= \left(\frac{\xi}{\sigma}J_C(\pi_k) + \frac{\xi}{\sigma(1-\gamma)}\mathbb{E}_{s \sim d^{\pi_k}, a \sim \pi_{k+1}}[A_C^{\pi_k}(s,a)] + 1\right)^{-\frac{1}{\xi}}\left(\frac{\xi}{\sigma(1-\gamma)}\mathbb{E}_{s \sim d^{\pi_k}, a \sim \pi_{k+1}}\left[\frac{2\gamma \epsilon_C^{\pi_{k+1}}}{1-\gamma}D_{TV}(\pi_{k+1}\|\pi_k)[s]] + 1\right)^{-\frac{1}{\xi}}
\end{aligned}
\tag{57}
$$

To simplify the representation, we denote:

$$
\begin{aligned}
\mathcal{J} &= \frac{\xi}{\sigma}\left(J_C(\pi_k) + \frac{1}{1-\gamma}\mathbb{E}_{s \sim d^{\pi_k}, a \sim \pi}[A_C^{\pi_k}(s,a)]\right) + 1 \\
\mathcal{E} &= \frac{\xi}{\sigma(1-\gamma)}\mathbb{E}_{s \sim d^{\pi_k}, a \sim \pi_{k+1}}\left[\frac{2\gamma \epsilon_C^{\pi_{k+1}}}{1-\gamma}D_{TV}(\pi_{k+1}\|\pi_k)[s]\right]
\end{aligned}
\tag{58}
$$

Then the constraint violation probability under EVO is less:

$$P(C > d) < (\mathcal{J}(\mathcal{E} + 1))^{-\frac{1}{\xi}} \tag{59}$$

The constraint optimization objective in expectation-based CRL is:

$$q_\mu = J_C < d \tag{60}$$

Then the constraint violation probability under expectation-based CRL is:

$$P(C > d) < 1 - \mu \tag{61}$$

EVO has a lower constraint violation probability than the expectation-based CRL by a margin of $\nu_0$:

$$1 - \mu - (1 - \mu - \nu_0) = \nu_0 \tag{62}$$

$\square$

### B.3. Comparison of Variance with Quantile Regression

**Lemma B.4.** *The asymptotic distribution theory ([Koenker & Bassett Jr, 1978](#)). Let $\{q_{\mu_1}, \cdots, q_{\mu_M}\}$ with $0 < \mu_1 < \cdots < \mu_M < 1$, denote a sequence of unique sample quantiles from random samples of size $N$ from a population with inverse distribution function $q_\mu = F_H^{-1}(\mu)$. If $F_H$ is continuous and has continuous and positive density $f_H$ at $q_{\mu_i}$, $i = 1, \cdots, M$, then the quantile regression estimator $q_\mu$ to the true quantile value $q_\mu^*$ converges in distribution to an Gaussian random vector with mean 0 and variance $\Omega$:*

$$\sqrt{N}(q_\mu - q_\mu^*) \to \mathcal{N}(0, \Omega) \tag{63}$$

*The variance $\Omega(\mu_1, \cdots, \mu_M)$ with typical element:*

$$\omega_{ij} = \frac{\mu_i(1 - \mu_j)}{f(q_{\mu_i})f(q_{\mu_j})} \tag{64}$$

**Theorem B.5** (Variance of EVO). *Let $q_{\frac{\nu}{1-\mu}}^H$ denote the quantile in GPD for random samples of size $N$ from a population with inverse distribution function $q_{\frac{\nu}{1-\mu}}^H = F_H^{-1}(\frac{\nu}{1-\mu})$. If the cumulative distribution function $F_H$ is continuous and has continuous and positive density $f_H$ at $q_{\frac{\nu}{1-\mu}}^H$, then the quantile estimator $q_{\frac{\nu}{1-\mu}}^H$ to the true quantile value $q_{\frac{\nu}{1-\mu}}^*$ converges in distribution to an Gaussian random vector with mean 0 and variance $\Omega$*

$$\sqrt{N}(q_{\frac{\nu}{1-\mu}}^H - q_{\frac{\nu}{1-\mu}}^*) \to \mathcal{N}(0, \Omega) \tag{65}$$

*The variance of EVO to estimate the quantile $\frac{\nu}{1-\mu}$ is:*

$$\Omega = \frac{\nu(1 - \mu - \nu)}{N(1 - \mu)^2 f_H^2(q_{\frac{\nu}{1-\mu}}^H)} \tag{66}$$

*where $\mu$ denotes the safety boundary quantile, $\nu$ denotes the exploitation range, and $N$ denotes the number of samples from the distribution that are used to estimate the quantile values. EVO has a lower variance than that in quantile regression methods:*

$$\Omega_2 = \frac{(\mu + \nu)(1 - \mu - \nu)}{N f_C^2(q_{\mu+\nu})} \tag{67}$$

*where $f_C$ denotes the probability density function.*

*Proof.* According to Lemma B.4, the variance of the estimator under the quantile level $\tau$ is:

$$\Omega = \frac{\tau(1-\tau)}{N f^2(\theta_\tau)} \tag{68}$$

where $N$ denotes the number of samples from the distribution that are used to estimate the quantile values.

The variance $\Omega_1$ of EVT to estimate the quantile $\frac{\nu}{1-\mu}$ is:

$$
\begin{aligned}
\Omega_1 &= \frac{(\frac{\nu}{1-\mu})(1 - \frac{\nu}{1-\mu})}{N f_H^2(q_{\frac{\nu}{1-\mu}}^H)} \\
&= \frac{\nu(1-\mu-\nu)}{N(1-\mu)^2 f_H^2(q_{\frac{\nu}{1-\mu}}^H)}
\end{aligned}
\tag{69}
$$

The variance $\Omega_2$ of quantile regression to estimate the quantile $\mu + \nu$ is:

$$\Omega_2 = \frac{(\mu+\nu)(1-\mu-\nu)}{N f_C^2(q_{\mu+\nu})} \tag{70}$$

where $f_C$ denotes the density of the overall cost distribution. According to Equation 10, $f_C$ is related to $f_H$:

$$f_C(q_\mu + z) = (1-\mu) f_H(z) \tag{71}$$

$$f_C(q_{\mu+\nu}) = (1-\mu) f_H\left(\frac{\nu}{1-\mu}\right) \tag{72}$$

Then we get the variance $\Omega_2$ of quantile regression is:

$$
\begin{aligned}
\Omega_2 &= \frac{(\mu+\nu)(1-\mu-\nu)}{N f_C^2(q_{\mu+\nu})} \\
&= \frac{(\mu+\nu)(1-\mu-\nu)}{N(1-\mu)^2 f_H^2(q_{\frac{\nu}{1-\mu}}^H)}
\end{aligned}
\tag{73}
$$

Thus the variance of EVO is lower than that of quantile regression: $\Omega_1 < \Omega_2$.

$\square$

## C. Experiment

### C.1. Additional Experiments

#### C.1.1. ADDITIONAL BASELINES

We conducted additional experiments across multiple tasks, comparing EVO with PPO-Lagrangian and RCPO (Tessler et al., 2018). PPO-Lagrangian is a primal-dual method that transforms constrained problems into unconstrained ones by introducing dual variables, and then optimizes the objective based on the PPO algorithm. RCPO proposes a multi-timescale Lagrangian method, which utilizes penalty signals to guide policy updates toward constraint satisfaction.

As shown in Figures 9 and 10, PPO-Lagrangian exhibits significant oscillations during training, and both PPO-Lagrangian and RCPO frequently violate constraints. In contrast, EVO consistently maintains constraint satisfaction across all tasks while achieving superior policy performance. In SafetyCarCircle1-v0, although RCPO achieves slightly higher returns, it exhibits substantial constraint violations, indicating that its policy is unsafe.

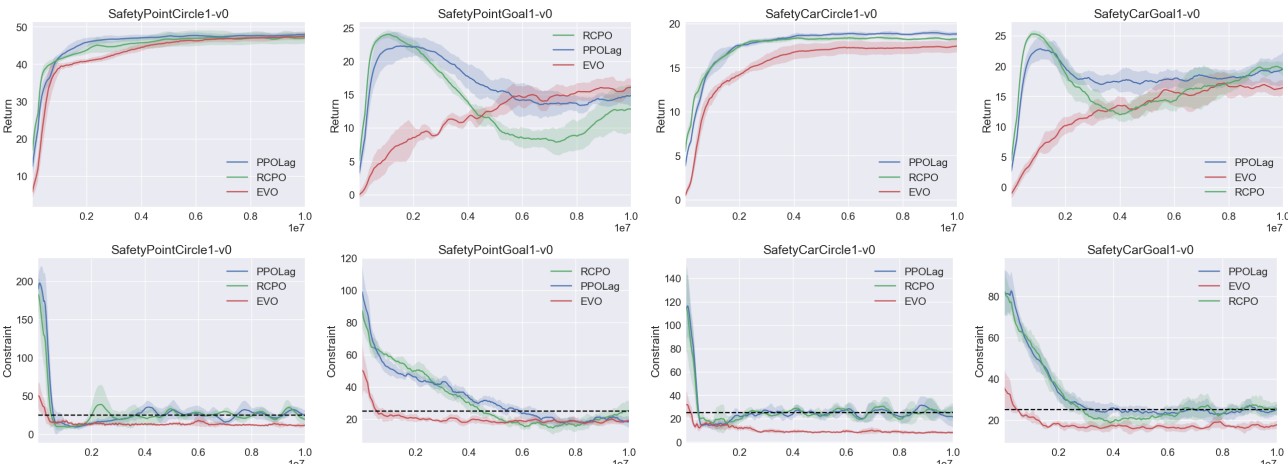

Figure 9: Comparison of EVO to PPO-Lagrangian and RCPO on Safety Gym. The x-axis is the total number of training steps, the y-axis is the average return or constraint. The solid line is the mean and the shaded area is the standard deviation. The dashed line is the constraint threshold which is 25.

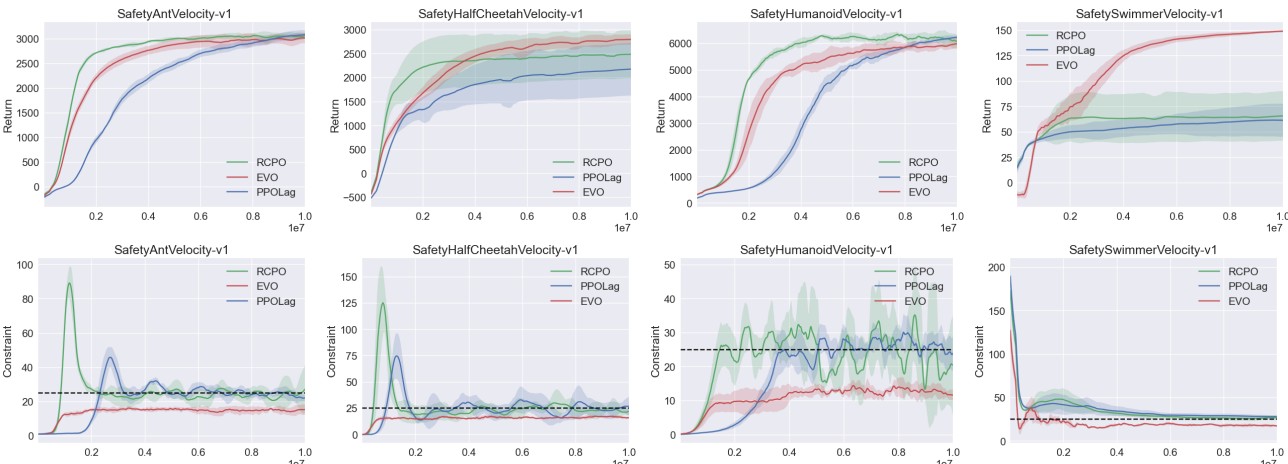

Figure 10: Comparison of EVO to PPO-Lagrangian and RCPO on Safety MuJoCo. The x-axis is the total number of training steps, the y-axis is the average return or constraint.

### C.1.2. TRAINING TIME

To evaluate the computational overhead of EVO, we conducted experiments across different environments, measuring both total training time and the time spent on GPD fitting. As shown in Table 1, EVO introduces only a limited increase in training time compared to CPO, and GPD fitting process is highly efficient, taking only a few seconds in total. This indicates that EVO introduces minimal computational overhead.

### C.2. Additional Results

In CRL algorithms, it is not sufficient to compare rewards or constraint violation rates in isolation. To quantitatively compare the overall performance of different algorithms in terms of both policy performance and constraint satisfaction during training, we design the following evaluation metric:

$$ratio = \frac{AverageReturn}{ConstraintViolationRate + \xi} \tag{74}$$

where $AverageReturn$ denotes the average return across all episodes during training, and $ConstraintViolationRate$ represents the proportion of episodes in which constraint violations occur. The constant $\xi$ is introduced to ensure numerical

| Environment | CPO | EVO(10 samples) | EVO(20 samples) | EVO(50 samples) | EVO(100 samples) |
|---|---|---|---|---|---|
| SafetyPointCircle1 | 11h 23m 28s | 11h 53m 19s (8s) | 11h 52m 28s (8s) | 11h 53m 31s (8s) | 11h 55m 22s (9s) |
| SafetyPointGoal1 | 11h 29m 42s | 11h 58m 51s (8s) | 11h 59m 47s (7s) | 11h 58m 33s (8s) | 11h 59 m 8s (9s) |

Table 1: EVO training time with different sample sizes compared to CPO. The time data includes the total training time for EVO, with the GPD fitting time shown in parentheses "()". EVO increases limited training time compared to CPO. GPD fitting takes only a few seconds in total.

stability. The final $ratio$ is then normalized. The corresponding results of $ratio$ are shown in Table 2. The results show that EVO achieves the best performance across multiple tasks, effectively balancing policy performance and constraint satisfaction.

| Environment | EVO (ours) | CPO | Saute | Simmer | WCSAC |
|---|---|---|---|---|---|
| SafetyCarGoal1 | **0.267** | 0.169 | 0.190 | 0.212 | 0.163 |
| SafetyCarCircle1 | **0.237** | 0.147 | 0.229 | 0.228 | 0.156 |
| SafetyPointGoal1 | **0.275** | 0.175 | 0.192 | 0.186 | 0.172 |
| SafetyPointCircle1 | **0.236** | 0.186 | 0.177 | 0.191 | 0.211 |
| SafetyAntVelocity | **0.250** | 0.132 | 0.245 | 0.244 | 0.129 |
| SafetyHalfCheetahVelocity | **0.233** | 0.166 | 0.231 | 0.224 | 0.146 |
| SafetyHumanoidVelocity | **0.219** | 0.168 | 0.213 | 0.214 | 0.186 |
| SafetySwimmerVelocity | 0.213 | 0.159 | **0.214** | 0.212 | 0.180 |

Table 2: The mean performance of $ratio$ across mutiple environments, with the bolded data indicating the maximum $ratio$.

## C.3. Environments

Our experimental environments consist of two types of tasks, navigation in Safety Gymnasium and velocity in Safety MuJoCo.

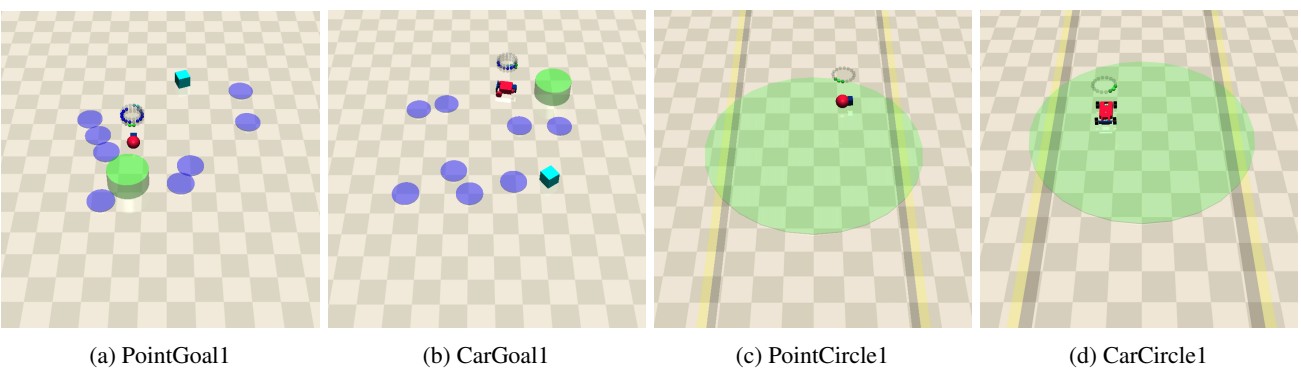

(a) PointGoal1      (b) CarGoal1      (c) PointCircle1      (d) CarCircle1

Figure 11: Environments in Safety Gym.

### C.3.1. SAFETY GYM

Figure 11 shows the environments in the Safety Gym. Safety Gym is the standard API for safe reinforcement learning developed by Open AI. The agent perceives the world through the sensors of the robots and interacts with the environment via its actuators in Safety Gym. In this work, we consider two agents, Point and Car, and two tasks, Goal and Circle.

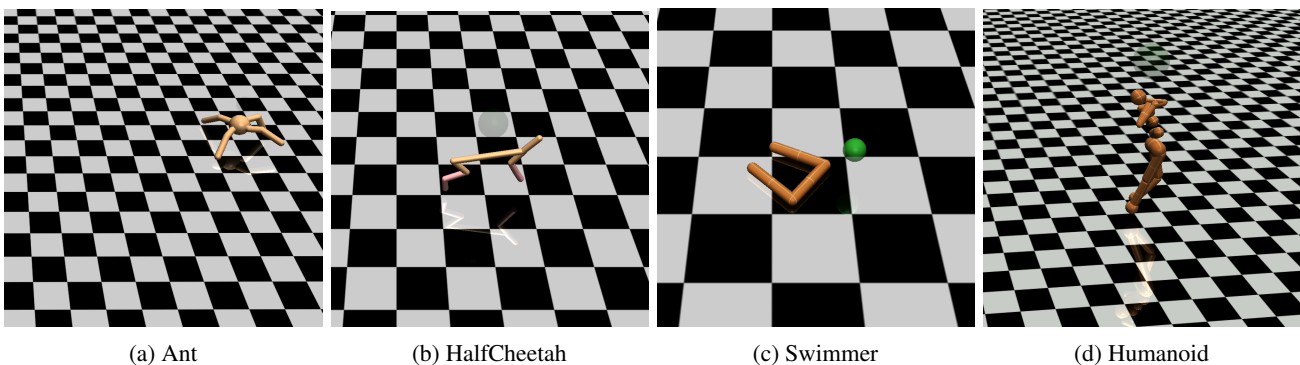

(a) Ant        (b) HalfCheetah        (c) Swimmer        (d) Humanoid

Figure 12: Environments in Safety MuJoCo.

The Point is a simple robot constrained to a two-dimensional plane. It is equipped with two actuators, one for rotation and another for forward/backward movement. It has a small square in front of it, making it easier to visually determine the orientation of the robot. The action space in Point consists of two dimensions ranging from -1 to 1, and the observation space consists of twelve dimensions ranging from negative infinity to positive infinity.

The Car is a more complex robot that moves in three-dimensional space and has two independently driven parallel wheels and a freely rotating rear wheel. For this robot, both steering and forward/backward movement require coordination between the two drive wheels. The action space of Car includes two dimensions with a range from -1 to 1, while the observation space consists of 24 dimensions with a range from negative infinity to positive infinity.

**Goal:** The agent is required to navigate towards the location of the goal. Upon successfully reaching the goal, the goal location is randomly reset to a new position while maintaining the remaining layout unchanged. The rewards in the task of Goal are composed of two components: reward distance and reward goal. In terms of reward distance, when the agent is closer to the Goal it gets a positive value of reward, and getting farther will cause a negative reward. Regarding the reward goal, each time the agent successfully reaches the Goal, it receives a positive reward value denoting the completion of the goal. In SafetyGoal1, the Agent needs to navigate to the Goal's location while circumventing Hazards. The environment consists of 8 Hazards positioned throughout the scene randomly.

**Circle:** Agent is required to navigate around the center of the circle area while avoiding going outside the boundaries. The optimal path is along the outermost circumference of the circle, where the agent can maximize its speed. The faster the agent travels, the higher the reward it accumulates. The episode automatically ends if the duration exceeds 500 time steps. When out of the boundary, the agent gets an activated cost.

### C.3.2. SAFETY MUJOCO

The agent in Safety MuJoCo is provided by OpenAI Gym, and it is trained to move along a straight line while constrained with a velocity limit. Figure 12 illustrates the different environments.

Velocity tasks are also an important class of tasks that apply RL to reality, requiring an agent to move as quickly as possible while adhering to velocity constraint. These tasks have significant implications in various domains, including robotics, autonomous vehicles, and industrial automation.

If velocity of current step exceeds the threshold of velocity, then receive an scalar signal 1, otherwise 0.

### C.4. Hyperparameters

The exploitation range $\nu$ is adaptively adjusted based on the safety and performance of the current policy $\pi$. The objective for the adaptive $\nu$ is:

$$\mathcal{L}(\nu) = -\nu \left( J_C(\pi) - d \right) \tag{75}$$

The gradient with respect to $\nu$ is:

$$\nabla_\nu \mathcal{L}(\nu) = - \left( J_C(\pi) - d \right) \tag{76}$$

Thus $\nu$ is updated as follows:

$$\nu \leftarrow clip(\nu - \alpha \nabla_\nu \mathcal{L}(\nu), 0) \tag{77}$$

where $\alpha$ denotes the learning rate, and the clip operation ensures that $\nu \geq 0$. If $J_C(\pi) > d$, $\nu$ increases, raising the quantile in GPD, which in turn reduces $J_C$ and promotes a safer policy. Conversely, if $J_C(\pi) < d$, $\nu$ decreases, lowering the quantile in GPD and encouraging exploration.

All experiments are implemented in Pytorch 2.0.0 and CUDA 11.3 and performed on Ubuntu 20.04.2 LTS with a single GPU (GeForce RTX 3090). The hyperparameters are summarized in Table 3.

| Parameter | EVO | Saute | Simmer | CPO | WCSAC |
|---|---|---|---|---|---|
| hidden layers | 2 | 2 | 2 | 2 | 2 |
| hidden sizes | 64 | 64 | 64 | 64 | 64 |
| activation | $tanh$ | $tanh$ | $tanh$ | $tanh$ | $tanh$ |
| actor learning rate | $1e^{-3}$ | $1e^{-3}$ | $1e^{-3}$ | $1e^{-3}$ | $1e^{-3}$ |
| critic learning rate | $1e^{-3}$ | $1e^{-3}$ | $1e^{-3}$ | $1e^{-3}$ | $1e^{-3}$ |
| batch size | 128 | 128 | 128 | 128 | 128 |
| trust region bound | 0.01 | $N/A$ | $N/A$ | 0.01 | $N/A$ |
| discount factor gamma | 0.99 | 0.99 | 0.99 | 0.99 | 0.99 |
| GAE gamma | 0.95 | 0.95 | 0.95 | 0.95 | 0.95 |
| normalization coefficient | $1e^{-3}$ | $1e^{-3}$ | $1e^{-3}$ | $1e^{-3}$ | $1e^{-3}$ |
| clip ratio | $N/A$ | 0.2 | 0.2 | $N/A$ | 0.2 |
| conjugate gradient damping | 0.1 | $N/A$ | $N/A$ | 0.1 | $N/A$ |
| initial lagrangian multiplier | $N/A$ | $1e^{-3}$ | $1e^{-3}$ | $N/A$ | $1e^{-3}$ |
| lambda learning rate | $N/A$ | 0.035 | 0.035 | $N/A$ | 0.035 |

Table 3: Hyperparameters

