# OpenReview forum: "Extreme Value Policy Optimization for Safe Reinforcement Learning"
_ICML.cc/2025/Conference — ICML 2025 poster_

### Official Review · Reviewer_o4Co · 2025-02-25

**Overall Recommendation:** 3

**Summary:**

The paper introduces Extreme Value policy Optimization (EVO), an algorithm that enhances safety in reinforcement learning by leveraging Extreme Value Theory (EVT) to model and exploit extreme reward and cost samples. EVO features an extreme quantile constraint to capture tail risks and an extreme prioritization mechanism to amplify learning signals from rare but impactful samples. Theoretically, EVO guarantees strict constraint satisfaction at a zero-violation quantile level and exhibits lower constraint violation probability and variance compared to existing methods. Extensive experiments validate EVO's effectiveness in reducing constraint violations while maintaining strong policy performance.

**Claims And Evidence:**

The claims made in the submission are generally well-supported by clear and convincing evidence.

**Essential References Not Discussed:**

N/A

**Experimental Designs Or Analyses:**

I did not find any obvious issues in the experimental designs or analyses in the paper.

**Methods And Evaluation Criteria:**

Yes, the paper's proposed methods and evaluation criteria are well-suited for the problem and application.

**Other Comments Or Suggestions:**

N/A

**Other Strengths And Weaknesses:**

### Strengths

The introduction of the extreme quantile constraint based on the Generalized Pareto Distribution (GPD) is a unique contribution. This mechanism allows the algorithm to explicitly model and exploit extreme samples, which is crucial for reducing constraint violations.

The paper provides strong theoretical guarantees on constraint satisfaction, violation probability, and variance reduction. These guarantees are essential for building trust in RL systems and ensuring their reliability.



### Weaknesses

The integration of EVT and the proposed mechanisms adds complexity to the algorithm. This might make it challenging to implement and tune for practitioners, especially those without a strong background in EVT. In contrast, WCSAC with a Gaussian approximation seems much easier to use.

The use of EVT relies on certain assumptions about the distribution of extreme events. While the paper demonstrates that these assumptions hold in the tested environments, they might not generalize to all real-world scenarios.

The paper assumes that sufficient extreme samples can be collected to fit the GPD accurately. In environments where extreme events are extremely rare, this might be a limiting factor.

**Questions For Authors:**

1 What are the minimum sample requirements for EVO to effectively fit the GPD and achieve reliable performance? How does the algorithm handle environments where extreme samples are very scarce?

2 Could the authors provide more details on the computational overhead associated with fitting the Generalized Pareto Distribution (GPD) and performing off-policy importance resampling? Specifically, how does the computational cost scale with the size of the dataset and the complexity of the environment?

**Relation To Broader Scientific Literature:**

Traditional CRL methods, such as Constrained Policy Optimization (CPO) (Achiam et al., 2017), focus on optimizing policies to ensure that the expected cost remains below a predefined threshold. However, these methods often fail to account for the variability in the cost distribution, especially in the tail, leading to frequent constraint violations. EVO addresses this limitation by explicitly modeling the tail behavior using EVT.

Methods like WCSAC (Yang et al., 2021) use probabilistic constraints and approximate the cost distribution with a Gaussian model to compute Conditional Value-at-Risk (CVaR). However, Gaussian approximations are inadequate for capturing the tail behavior accurately. EVO improves upon this by using the Generalized Pareto Distribution (GPD) to model the tail, providing a more accurate representation of extreme events.

**Theoretical Claims:**

I did not find any obvious issues in the proofs for theoretical claims in the paper.

---

> ### Author Rebuttal · Authors · 2025-03-31
>
> We sincerely appreciate the reviewer's positive and insightful comments. The following are the detailed responses to the points raised by Reviewer o4Co.
>
>
> >The integration of EVT and the proposed mechanisms adds complexity to the algorithm. This might make it challenging to implement and tune for practitioners, especially those without a strong background in EVT.
>
> **Response:**
> We appreciate the reviewer’s insightful comment.
>
> To facilitate rapid implementation and support ease of use, we have open-sourced our code along with parameter settings. Moreover, EVO introduces almost no additional hyperparameters, greatly simplifying the tuning process.
>
> We thank the reviewer's insightful suggestions, and we plan to further refine and package EVO into a user-friendly library, enabling practitioners to apply our method via a simple function call without requiring deep knowledge of EVT.
>
>
> >What are the minimum sample requirements for EVO to effectively fit the GPD and achieve reliable performance? How does the algorithm handle environments where extreme samples are very scarce?
>
> **Response:**
> We appreciate the reviewer’s valuable comments.
>
> We conducted experiments with varying sample sizes (10, 20, 50, and 100) to evaluate the minimum number of samples required for EVO.
> As shown in Figure 1 (<https://anonymous.4open.science/r/11409-C857/experiments4.pdf>), larger sample sizes generally lead to improved constraint satisfaction. Specially, in SafetyPointCirclel-v0, EVO maintains strong constraint satisfaction and performance even with limited 10 samples. In SafetyPointGoal1-v0, constraint satisfaction is consistently achieved once sample size exceeds 20
> Overall, a minimum of 20 samples is sufficient for achieve reliable performance in EVO.
> Additionally, in our main experiments, with the same sample size of 20, EVO demonstrates superior constraint satisfaction and policy performance compared to other baselines, indicating its effectiveness even with limited samples.
>
> To address the challenge of very scarce extreme-value samples, EVO augments extreme samples using off-policy samples and apply importance resampling to correct the distribution shift. Furthermore, in special cases where extreme samples are nearly absent or difficult to collect, offline extreme samples can be provided in advance to ensure GPD fitting remains feasible.
>
>
>
> >Could the authors provide more details on the computational overhead associated with fitting the Generalized Pareto Distribution (GPD) and performing off-policy importance resampling? Specifically, how does the computational cost scale with the size of the dataset and the complexity of the environment?
>
> **Response:**
> We appreciate the reviewer’s valuable comments.
>
> To evaluate the computational overhead of EVO, we measured total training time for both EVO and CPO across multiple environments, along with the time spent on GPD fitting. As shown in Table 1 (<https://anonymous.4open.science/r/11409-C857/experiments4.pdf>), EVO only adds a limited training time compared to CPO, and GPD fitting is highly efficient, taking only a few seconds in total. This indicates that EVO introduces minimal computational overhead.
>
> We also assessed the impact of dataset size by training EVO with varying sample sizes. The results show that increasing the sample size has negligible effect on overall training time.
>
> To evaluate scalability of EVO with environmental complexity, we compared training times across environments of varying difficulty. While more complex environments naturally require more time, EVO’s additional overhead remains comparable to that of CPO.

---

### Official Review · Reviewer_hnao · 2025-03-12

**Overall Recommendation:** 3

**Summary:**

The paper proposes Extreme Value Policy Optimization (EVO), a novel algorithm for safe reinforcement learning (RL) that addresses rare but high-impact extreme events in constrained RL (CRL). Traditional CRL methods optimize expected cumulative costs (e.g., $J_C(\pi) \leq d$), which overlook tail risks (e.g., "black swan" events). EVO integrates Extreme Value Theory (EVT) to model tail distributions of costs/rewards using Generalized Pareto Distributions (GPDs).

**Claims And Evidence:**

Supported by experiments (e.g., Figure 3-4) and Theorem 4.1-4.3. However, comparisons to non-EVT quantile methods (e.g., VaR) are missing in ablation studies.

Theorem 4.3 shows $\Omega < \Omega_2$ (EVO vs. quantile regression), but bias in GPD parameter estimation (e.g., $\xi$, $\sigma$) is not addressed.

**Essential References Not Discussed:**

No

**Experimental Designs Or Analyses:**

Figure 6 shows variance reduction with resampling but lacks metrics like convergence speed.

**Methods And Evaluation Criteria:**

EVT-based tail modeling is novel but sensitive to threshold selection for $q_\mu$. The prioritization mechanism ($p = \omega_r + \omega_c$) is intuitive but assumes accurate GPD fits.

Tasks like SafetyCarCircle1 are standard but lack dynamic/adversarial scenarios to test robustness.

**Other Comments Or Suggestions:**

No

**Other Strengths And Weaknesses:**

**Strengths**:

 First to integrate EVT with CRL for tail risk mitigation.

Theorems 4.1-4.3 and ablation studies validate contributions.

 Zero violations in experiments are critical for real-world safety.

**Weaknesses**:

EVT assumes i.i.d. extremes, which may fail in non-stationary RL.

GPD fitting and prioritization add complexity; training time is not quantified.

**Questions For Authors:**

1.  How does EVO ensure policies $\pi_{k+1}$ and $\pi_k$ adhere to the quantile-based objective with off-policy data in Theorem 4.1?
2.  How is $q_\mu$ initialized/updated? Could biased thresholds harm GPD fits?
3.  Why omit PPO-Lagrangian/RCPO?
4.  Does EVO require more samples to collect extremes?

**Relation To Broader Scientific Literature:**

EVO advances CRL by integrating EVT for tail risks, contrasting expectation-based (CPO) and Gaussian-based (WCSAC) methods. Connections to distributional RL (e.g., Bellemare et al., 2017) are underexplored.

**Theoretical Claims:**

- **Theorem 4.3**: Variance reduction is valid but ignores bias in $\xi$ estimation, which affects $q^{H}_{\frac{\nu}{1-\mu}}$.

---

> ### Author Rebuttal · Authors · 2025-03-31
>
> We sincerely appreciate the reviewer's positive and insightful comments. The following are the detailed responses to the points raised by Reviewer hnao.
>
> >Comments regarding the accuracy of GPD fitting and the bias in GPD parameter estimation.
>
> **Response:**
> We appreciate the reviewer’s insightful comments and agree that parameter estimation bias exists. In our work, we use maximum likelihood estimation for GPD parameters, which inevitably introduces bias. This bias is inherent in statistical distribution fitting and not unique to our method.
>
> To assess the impact of bias, we saved samples during training and fitted both GPD and Gaussian distributions. Using the Kolmogorov–Smirnov test to measure fitting accuracy (lower values indicate better fit), we found that GPD consistently provides accurate fits across various data distributions despite with estimation bias, as shown in Figure 1 (<https://anonymous.4open.science/r/11409-C857/experiments3.pdf>).
>
> >Comparisons to non-EVT quantile methods are missing in ablation studies.
>
> **Response:**
> We conducted an ablation study on the EVT-based quantile, as shown by the green curves in Figure 5(a)(b) in the paper. The results show that removing it reduces policy performance, demonstrating the advantage of explicitly modeling extremes in EVO.
>
> >Questions regarding the selection and update of $q_\mu$.
>
> **Response:**
> In this paper, the quantile $q_\mu$ represents the safety boundary reflecting the overall constraint distribution. Rather than assuming $q_\mu$ equals the expectation, we explicitly set it to the expected constraint value under the current policy. The corresponding quantile level $\mu$ is then derived from the constraint distribution at each update.
>
> >Figure 6  lacks metrics like convergence speed.
>
> **Response:**
> We provide the learning curves corresponding to Figure 6, with off-policy importance resampling ablated. As shown in Figure 2 (<https://anonymous.4open.science/r/11409-C857/experiments3.pdf>), constraint satisfaction in EVO converges after $1 \times 10^6$ steps.
>
>
> >Omitting sensitivity analysis for $\nu$ adaptation.
>
> **Response:**
> We constructed experiments to evaluate the sensitivity of different $\nu$, as shown in Figure 3 (<https://anonymous.4open.science/r/11409-C857/experiments3.pdf>). The results show that  EVO is robust to the initial choice of $\nu$, as it is adaptively updated during training based on current policy performance, as shown in Appendix C.4.
>
>
> >Connections to distributional RL are under-explored.
>
> **Response:**
> We will include a more detailed discussion on distributional RL in related work.
>
> While distributional RL models the full return distribution instead of the expectation, it does not address constraints. WCSAC expands distributional RL into constrained RL by introducing a distributional safety critic based on CVaR.
>
> >EVT may fail to non-i.i.d. extremes.
>
> **Response:**
> For non-i.i.d. data, methods such as Block Maxima or clustering modeling can be applied firstly to make the extreme samples approximately i.i.d., thereby enabling the effective application of EVT.
>
>
> > Training time is not quantified.
>
> **Response:**
> We conducted experiments across different environments, measuring both total training time and the time spent on GPD fitting. As shown in Table 1 (<https://anonymous.4open.science/r/11409-C857/experiments3.pdf>), EVO increases limited training time compared to CPO, indicating minimal computational overhead.
>
> We also evaluate the impact of sample size and found that the additional time remains minimal as the sample size increases. The GPD fitting is highly efficient, taking only a few seconds in total, with negligible overhead even at larger sample sizes.
>
>
> >How ensuring policies $\pi_{k+1}$ and $\pi_k$ adhere to the quantile-based objective with off-policy data in Theorem 4.1?
>
> **Response:**
> For off-policy data, we apply an importance resampling method, as shown in Equation (18) of the paper, to correct the distributional discrepancy between the current policy and the off-policy samples.
>
>
> >Why omit PPO-Lagrangian/RCPO?
>
> **Response:**
> We conducted additional experiments across multiple tasks, comparing EVO with PPO-Lagrangian and RCPO. As shown in Figure 4 and Figure 5 (<https://anonymous.4open.science/r/11409-C857/experiments3.pdf>), PPO-Lagrangian exhibits significant oscillation during training, and both PPO-Lagrangian and RCPO frequently violate constraints. In contrast, EVO consistently maintains constraint satisfaction across all tasks while achieving superior policy performance. In SafetyCarCircle1-v0, although RCPO achieves slightly higher returns, it exhibits substantial constraint violations, indicating that its policy is unsafe.
>
>
> >Does EVO require more samples to collect extremes?
>
> **Response:**
> EVO does not require more samples to collect the extremes. In all experiments its using the same size of samples as the other baselines.

---

### Official Review · Reviewer_bFMW · 2025-03-15

**Overall Recommendation:** 3

**Summary:**

This paper presents the Extreme Value policy Optimization (EVO) algorithm for safe reinforcement learning. It integrates Extreme Value Theory (EVT) to model and utilize extreme samples. EVO introduces an extreme quantile constraint and an extreme prioritization mechanism. Theoretically, it has a lower constraint violation probability and variance than baselines. Experiments in Safety Gymnasium and MuJoCo show that EVO reduces constraint violations while maintaining competitive policy performance.

**Claims And Evidence:**

Most of the claims in the paper are supported by clear and convincing evidence.

**Essential References Not Discussed:**

n/a

**Experimental Designs Or Analyses:**

The paper is reasonable and effective in experimental design and analysis

**Methods And Evaluation Criteria:**

The methods and evaluation criteria proposed in the paper are consistent with the research problems.

**Other Comments Or Suggestions:**

-	In the right column of line 399, there is a typo in“Figure 5a and 5a”.

**Other Strengths And Weaknesses:**

See other parts.

**Questions For Authors:**

See other parts.

**Relation To Broader Scientific Literature:**

n/a

**Theoretical Claims:**

I check the theoretical claims in the main text. I have some confusions with the theoretical parts:

- In lines 210-211, the conditional probability does not follow the GPD. According to theorem 3.1, it only follows GPD as $q_\mu \to \infty$. The following argument should establish in approximately way.
- Why (8) holds? It seems that $Z \le z \iff C – q_{\mu} \le q_{\mu + \nu} – q_{\mu} \iff C \le q_{\mu + \nu}$.
- In Theorem 4.1, I cannot understand this sentence: “$\pi_{k+1}, \pi_{k}$ are related by quantile-based constraint objective 11 .”
- The metrics investigated in section 4.4 should be defined formally and discussed more. Now It is confused that why these theorems demonstrate the advantage of EVO.

---

> ### Author Rebuttal · Authors · 2025-03-31
>
> We sincerely appreciate the reviewer's positive and insightful comments. The following are the detailed responses to the points raised by Reviewer bFMW.
>
> >In lines 210-211, the conditional probability does not follow the GPD. According to theorem 3.1, it only follows GPD as $q_\mu \to \infty$. The following argument should establish in approximately way.
>
> **Response:**
> We appreciate the reviewer’s insightful comment and agree with the reviewer’s perspective. According to extreme value theory, extreme value samples asymptotically follow GPD. Therefore, we use the asymptotic equality symbol to reflect this relationship, as shown in Equations (8) and (9) in our paper.
>
> Empirically, our experiments show that GPD provides a good fit for modeling extreme value samples. To validate this, we collected training data from multiple environments and fitted both GPD and Gaussian distributions. Furthermore, we employed the Kolmogorov-Smirnov test to quantify the fitting accuracy, where lower scores indicate better fit. As shown in Figure 1 (<https://anonymous.4open.science/r/11409-C857/experiments2.pdf>), GPD presents accurate fitting performance across various data distributions, and it consistently outperforms the Gaussian distribution in capturing tail behavior.
>
> We also acknowledge that in special cases, such as when the difference between extreme and normal values is small, GPD may not accurately capture the extremes. To address this issue, nonlinear transformations or similar methods can be applied to amplify the difference between extreme and normal values, and then fit the data with GPD for improved precision.
>
>
> >Why (8) holds? It seems that $Z \le z \iff C – q_{\mu} \le q_{\mu + \nu} – q_{\mu} \iff C \le q_{\mu + \nu}$.
>
> **Response:**
> We appreciate the reviewer’s valuable comment.
> In this paper, the $z$ denotes the excess value beyond $q_\mu$, which contains the condition $z = q_{\mu + \nu} - q_\nu > 0$, as shown in Eq (12) in the paper. So $P(Z \le z) = P(C-q_\mu \le z|z>0) = P(C-q_\mu \le z| z>0)$, and $P(Z\le z) \neq P(C \le q_{\mu + \nu})$ .
>
> In this paper, we use the GPD $P(Z)$ to separately model the portion $Z$ of the overall distribution $C$ that exceeds $q_\mu$. According to EVT, $P(C - q_\mu \le z|C>q_\mu)$ asymptotically follows GPD, thus $P(C - q_\mu \le z|C>q_\mu) \backsimeq  P(Z \le z)$.
>
> According to the Eq. (7) in the paper:
> \begin{equation}
> \begin{aligned}
>     F_C(q_{\mu+\nu}) = P(C \le q_\mu + z) = P(C \le q_\mu) + P(C >q_\mu) P(C - q_\mu \le z|C>q_\mu)
> \end{aligned}
> \end{equation}
>
> Then we can get Eq (8):
> \begin{equation}
> \begin{aligned}
>     F_C(q_{\mu+\nu}) = P(C \le q_\mu + z) = \mu + \nu \backsimeq \mu + (1-\mu)P(Z\le z)
> \end{aligned}
> \end{equation}
>
> We appreciate the reviewer’s valuable comment for enhancing the clarity of our paper.
>
>
> >In Theorem 4.1, I cannot understand this sentence:"$\pi_{k+1}, \pi_{k}$ are related by quantile-based constraint objective 11."
>
> **Response:**
> We appreciate the reviewer’s valuable comment.
> This means that policy $\pi_{k+1}$ is obtained by optimizing policy $\pi_k$ according to objective function (11). Here, we follow the notation and description adopted in constrained RL works such as CPO and PCPO.
>
>
> >The metrics investigated in section 4.4 should be defined formally and discussed more. Now It is confused that why these theorems demonstrate the advantage of EVO.
>
> **Response:**
> We appreciate the reviewer’s valuable comments.
> Theorem 4.1 is about the constraint expectation, demonstrating that the updated policy in EVO has overall lower constraints compared to expectation-based CRL methods, which is empirically validated in multiple environments in Figures 3 and 4 of the paper.
>
> Theorem 4.2 focuses on the constraint violation probability, showing that EVO has a lower constraint violation probability during training compared to expectation-based CRL methods, as validated in Figures 3 and 4 in the paper across multiple environments.
>
> Theorem 4.3 is regarding the variance of extreme value estimation, proving that EVO provides more stable extreme value estimates compared to quantile regression methods, which is verified by the results in Figure 6 of the paper.
>
> We hope these discussions help clarify the theorems in Section 4.4.
>
>
>
> > In the right column of line 399, there is a typo in "Figure 5a and 5a".
>
> **Response:**
> We greatly appreciate the reviewer’s careful review and will correct this typo in the revised manuscript.

---

### Official Review · Reviewer_csFL · 2025-03-17

**Overall Recommendation:** 3

**Summary:**

The authors propose a novel approach, Extreme Value Policy Optimization (EVO), to handle rare but high-impact extreme value events by using the Extreme Value Theory (EVT). The EVO introduces an extreme quantile optimization objective and an extreme prioritization mechanism. Extensive experiments are conducted to demonstrate the effectiveness of EVO in terms of the reduction in constraint violations while maintaining the performance.

**Claims And Evidence:**

The claims made in the submission are generally supported by clear and convincing evidence.

**Essential References Not Discussed:**

No

**Experimental Designs Or Analyses:**

The experimental designs are sound and valid.

**Methods And Evaluation Criteria:**

The methods and evaluation criteria in the paper are aligned with the target problem.

**Other Comments Or Suggestions:**

No.

**Other Strengths And Weaknesses:**

Strength: The paper uses EVT to constrained RL to capture rare but high-impact extreme value events that previous methods overlook.
Weakness: The paper depends on the assumption that the extreme value events follow Generalized Pareto Distributions, which may not be true.

**Questions For Authors:**

Could the authors provide more details about the sample size for EVO to fit the GPD?

**Relation To Broader Scientific Literature:**

The paper addresses the limitation of expectation-based constrained reinforcement learning (Achiam et al 2017). EVO makes the improvement by explicitly modeling extreme events by EVT.

**Theoretical Claims:**

The proofs for theoretical claims are correct.

---

> ### Author Rebuttal · Authors · 2025-03-31
>
> We sincerely appreciate the reviewer's positive and insightful comments. The following are the detailed responses to the points raised by Reviewer csFL.
>
> >The paper depends on the assumption that the extreme value events follow Generalized Pareto Distributions, which may not be true.
>
> **Response:**
> We appreciate the reviewer’s insightful comments.
> According to extreme value theory, as extreme values increase, these samples asymptotically follow the GPD, which is proven to be effective for modeling extreme events in prior studies [1][2].
>
> To evaluate the fitting accuracy of GPD in our experiments, we collected training data across multiple environments and fitted both GPD and Gaussian distributions. Furthermore, we employed the Kolmogorov-Smirnov (KS) test to quantify the accuracy of the GPD and Gaussian fits, where lower values indicate more accurate fits. As shown in Figure 1 (<https://anonymous.4open.science/r/11409-C857/experiments1.pdf>), GPD presents accurate fitting performance across various data distributions. And GPD provides a more accurate fit for extreme samples than the Gaussian distribution and better captures the tail behavior of the distribution.
>
> We also acknowledge that in special cases, such as when the distinction between extreme and normal values is small, GPD may not provide a satisfactory fit. To address this issue, we can first apply methods like nonlinear transformations to amplify the difference between extreme and normal values, and then fit the data with GPD for improved accuracy.
>
> We thank the reviewer’s constructive feedback and view this as a promising direction for extending EVO to more practical applications in future research.
>
>
> >Could the authors provide more details about the sample size for EVO to fit the GPD?
>
> **Response:**
> We appreciate the reviewer’s valuable comment regarding the sample size in EVO.
> We conducted additional experiments varying the sample size used for GPD fitting and evaluated the corresponding policy performance. As shown in Figure 2 (<https://anonymous.4open.science/r/11409-C857/experiments1.pdf>), larger sample sizes generally lead to improved constraint satisfaction. Notably, in SafetyPointCircle1-v0, EVO maintains strong constraint satisfaction and performance even with limited 10 samples. In SafetyPointGoal1-v0, constraint satisfaction is consistently achieved once the sample size exceeds 20.
>
> In our main experiments, with the same samples size of 20, EVO demonstrates superior constraint satisfaction and policy performance compared to other baselines, indicating that it remains effective even with relatively small sample sets.
>
>
> [1] NS, K. S., Wang, Y., Schram, M., Drgona, J., Halappanavar, M., Liu, F., and Li, P. Extreme risk mitigation in reinforcement learning using extreme value theory. arXiv preprint arXiv:2308.13011, 2023.
>
> [2] Siffer, A., Fouque, P.-A., Termier, A., and Largouet, C. Anomaly detection in streams with extreme value theory. In Proceedings of the 23rd ACM SIGKDD international conference on knowledge discovery and data mining, pp. 1067–1075, 2017.

---

### Decision · Program_Chairs · 2025-05-01

**Decision:**

Accept (poster)

**Comment:**

This paper investigates the Constrained Reinforcement Learning problem and proposes the Extreme Value policy Optimization (EVO) algorithm, providing both theoretical and empirical evidence to demonstrate its superiority over existing baselines.

All reviewers acknowledge the novelty of introducing Extreme Value Theory (EVT) into constrained RL to effectively capture rare events. Although some assumptions in the paper—such as those related to the distribution of rare events and the additional computational cost—were questioned, these concerns were thoroughly addressed during the rebuttal period.

The authors clarified the assumptions used in the paper, including the rationale for modeling extreme events with the Generalized Pareto Distribution (GPD). Additionally, they provided experimental details regarding the computational overhead involved in fitting the GPD. The authors also addressed issues related to the proof details. In summary, the authors’ responses adequately resolved the concerns raised by the reviewers.

In light of the reviewers’ assessments following the rebuttal, this paper is suitable for acceptance.